# Tensor Train Diffusion: A Fast Solver for High-Dimensional Sampling

## Abstract

Diffusion models offer a powerful framework for sampling from complex probability densities by learning to reverse a noising process. A common approach involves solving for the time-reversed stochastic differential equation (SDE), which requires the score function of the evolving sample distribution. The logarithm of this distribution's density is governed by a Hamilton-Jacobi-Bellman (HJB) type partial differential equation (PDE). However, current methods for solving this PDE, such as PINNs or trajectory-based techniques, often suffer from long training times and significant sensitivity to hyperparameter tuning. In this work, we introduce a novel and efficient solver for the underlying HJB equation based on the functional tensor train (FTT) format. The FTT representation leverages latent low-rank structures to efficiently approximate high-dimensional functions, enabling both model compression and rapid computation. By integrating this efficient representation with a backward-in-time iterative scheme derived from backward stochastic differential equations (BSDEs), we develop a fast, robust and accurate sampling method. Our approach overcomes primary bottlenecks of existing techniques, enabling high-fidelity sampling from challenging target distributions with improved efficiency.

## 1 Introduction

Sampling from a complex, high-dimensional probability density

$$p_{\text{target}} = \frac{\rho_{\text{target}}}{\mathcal{Z}}, \tag{1}$$

which is specified only up to the normalizing constant $\mathcal{Z}$, constitutes a central challenge in modern machine learning, statistics, and the physical sciences. An especially versatile and powerful class of methods for this task is based on dynamical measure transport, which generates samples from the target distribution by evolving initial samples from a simple reference distribution – such as a standard Gaussian – along trajectories defined by stochastic or ordinary differential equations. One of the most successful paradigms within this framework is the time reversal of a noising SDE, forming the foundation of *diffusion models* (Ho et al., 2020; Song et al., 2021). In this approach, a forward SDE – typically chosen as an *Ornstein-Uhlenbeck process* – progressively transforms the target distribution into a simple prior by injecting noise over time. The central insight is that reversing this noising process yields a generative SDE that transports samples from the prior back to the target distribution. A classical result (Anderson, 1982) establishes that the drift of this time-reversed SDE is determined by the *score function*, $\nabla \log p(x, t)$, where $p(\cdot, t)$ denotes the density of the forward process at time $t$. Thus, the sampling problem reduces to accurately approximating this time-dependent score function.

Since the densities of SDEs are governed by the *Fokker-Planck equation*, one can employ the *Hopf-Cole* transform to relate the evolution of the log-density to a *Hamilton-Jacobi-Bellman* (HJB) equation (Berner et al., 2024). However, solving this PDE is notoriously challenging, particularly in the high-dimensional settings typical in practical applications, which are intractable for traditional mesh-based approaches. In the context of diffusion models, there exist attempts to approximate solutions to such HJB equations with neural networks, for example via *physics-informed neural networks* (PINNs) (Sun et al., 2024; Shi et al., 2024b). However, the optimization of neural networks with stochastic gradient descent (SGD) requires a large number of (potentially costly) evaluations of $\rho_{\text{target}}$ and relies on automatic differentiation to compute the higher-order derivatives appearing in

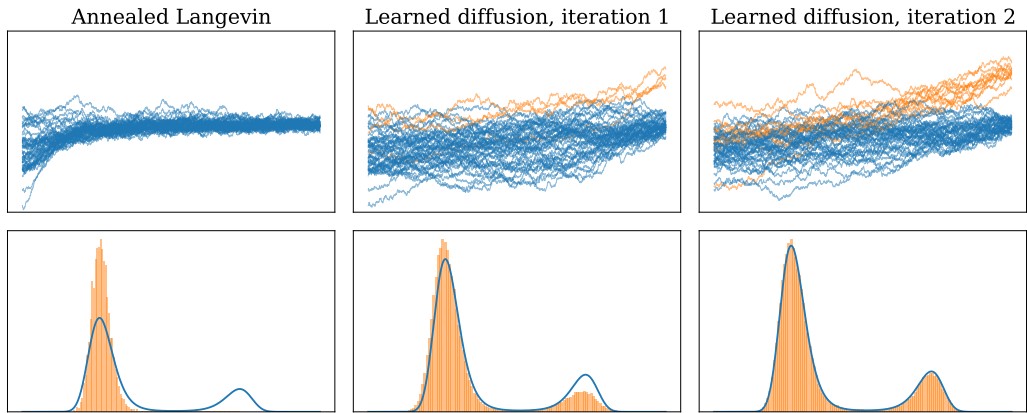

Figure 1: Overview of the proposed TTD method. From left to right: Annealed Langevin dynamics serve as an initialization for our approach. By learning the score function along relevant trajectories, TTD iteratively refines the sampling process. As a result, new modes are discovered and the quality of samples improves. The histograms illustrate the terminal samples of the trajectories in comparison to the target density.

the PDE. This leads to long training times, sensitivity to hyperparameters, and convergence to local minima (Krishnapriyan et al., 2021; Shi et al., 2024a; Xu et al., 2025).

In this work, we depart from neural network-based PDE solvers and propose a novel approach based on *functional tensor train* (FTT) representations. Tensor trains are particularly well-suited for this setting, as they can efficiently represent high-dimensional functions by exploiting latent low-rank structures, enabling both substantial model compression and fast computation. Moreover, the tensor train format is naturally compatible with backward-in-time regression schemes, which can be derived from the connection between HJB equations and backward stochastic differential equations (BSDEs). This yields an efficient implementation that circumvents the instabilities and complexities associated with SGD-based optimization. Our main contributions can be summarized as follows:

- We introduce *Tensor Train Diffusion* (TTD)[1], a novel solver for HJB equations that leverages connections between diffusion-based sampling, HJB equations, and BSDEs to achieve a favorable trade-off between computational cost and accuracy.
- We demonstrate the flexibility of TTD by incorporating different basis functions – including Legendre polynomials, B-splines, and Fourier bases – and substantially improve robustness compared to previous tensor-train-based PDE solvers, effectively handling sampling trajectories outside the primary training domain.
- We show that TTD can be used to sample from complex, multimodal target distributions of varying dimensionality, outperforming existing diffusion-based samplers in both speed and accuracy.

## 1.1 RELATED WORK

**Sampling from unnormalized densities.** Drawing samples from a density, specified up to the normalizing constant, is a crucial task in a wide range of computational sciences, ranging from molecular dynamics, to lattice field theory, to Bayesian statistics (Gelman et al., 2013; Zhang et al., 2023). Due to the great interest, a large amount of literature focused on developing corresponding sampling methods, often based on variants of *Markov chain Monte Carlo* (MCMC) and *Langevin dynamics* (Neal, 2001; Chopin, 2002; Del Moral et al., 2006). However, such algorithms typically require substantial tuning and long runtimes to sample from high-dimensional distributions with well-separated modes (Latuszyński et al., 2025; Brooks et al., 2011).

**Diffusion-based samplers.** To combat these issues, variational inference methods propose to turn the sampling problem into an optimization problem over a tractable family of distributions. Translating the success of diffusion models to sampling problems, recent works proposed to use parametrized SDEs for these families (Richter & Berner, 2024; Berner et al., 2024; Vargas et al.,

---

[1]Our code will be released upon acceptance and constitutes the first GPU-accelerated code of tensor trains written purely in PyTorch.

2024). Since the drifts of the SDEs are typically parametrized by neural networks that incorporate the gradient of the target density, such approaches can be viewed as a learnable corrections to Langevin dynamics, aiming to converge after finite trajectory lengths. However, this also leads to a significant amount of (potentially costly) density evaluations during SDE simulations, which are also needed for training. While such methods can outperform classical methods in terms of inference performance and efficiency, they suffer from a high upfront cost for optimizing the parameters of the SDE.

**Neural PDE solver.** Since the dynamics of SDEs are governed by Fokker-Planck equations, we can equivalently solve the corresponding PDE in order to optimize the variational family of SDEs. However, such high-dimensional PDEs are out of scope for traditional mesh-based methods, such as finite differences or finite elements, which suffer from the *curse of dimensionality*. There is a series of works proposing variants of physics-informed neural networks for this problem (Shi et al., 2024b; Albergo & Vanden-Eijnden, 2025; Sun et al., 2024). While they can scale to higher dimensions, they do not address the high training costs due to their reliance on evaluations of the target density and higher-order derivatives as well as high sensitivity to hyperparameters.

**Tensor Trains.** Many target densities of interest exhibit an inherent low-rank structure. To exploit this, we propose to approximate the drift using functional Tensor Trains (FTT, Oseledets (2013)) in their numerical realization, the *extended (functional) Tensor Train* format (Eigel et al., 2022; Strössner et al., 2024) is sometimes referred as *spectral tensor trains* (Bigoni et al., 2016). The effectiveness of such representations has already been demonstrated in related contexts, for example in sampling problems connected to HJB equations (Gruhlke et al., 2025), as well as for the Fokker–Planck and HJB equations in Dolgov et al. (2012). Low-rank tensor formats enable efficient regression-based solvers with broad applicability; see Bachmayr (2023) for an overview. Moreover, they constitute a rigorous nonlinear approximation class, with detailed analyses provided in Kazeev & Schwab (2018); Ali & Nouy (2020a;b; 2021); Bachmayr et al. (2021); Griebel et al. (2023); Bachmayr (2023). In this work, we build on the ideas developed in Richter et al. (2021; 2024) and extend them to the sampling setting. Inspired by Bouchard & Touzi (2004); Gobet et al. (2005); Huré et al. (2020), we notice that the Fokker-Planck equation is the linearization of a Hamilton-Jacobi-Bellman equation under the Hopf-Cole transform, such that the optimal drift can be written in terms of backward SDEs (BSDEs) that lead to simple regression problems. Together with the TT parametrization, we show that this reformulation of the PDE leads to an efficient and scalable sampler.

**Notation.** We define time-inversion as $\overleftarrow{f}(t) := f(T - t)$. For other notation, including our tensor notation, we refer to Table 1 in the appendix.

## 2 DIFFUSION-BASED SAMPLING

In this section, we demonstrate how learned stochastic processes can be employed to sample from the target distribution and introduce a stochastic algorithm that formalizes the associated learning task.

### 2.1 TIME-REVERSAL AND THE RELATED HAMILTON-JACOBI-BELLMAN PDE

Our sampling algorithm for $p_{\text{target}}$ builds on the idea of reversing a noising process that approximately evolves towards an easy-to-sample distribution $p_{\text{prior}}$ at terminal time $T$ (Song et al., 2021; Berner et al., 2024). To this end, we consider the process

$$\mathrm{d}Y_s = -\overleftarrow{f}(Y_s, s)\,\mathrm{d}s + \overleftarrow{\sigma}(s)\,\mathrm{d}W_s, \quad Y_0 \sim p_{\text{target}}, \tag{2}$$

where $W$ is standard Brownian motion. For suitable choices of $f \in C(\mathbb{R}^d \times [0, T], \mathbb{R}^d)$ and $\sigma \in C([0, T], \mathbb{R}^{d \times d})$ (e.g. $f(x, s) = x$, $\sigma(s) = \sqrt{2}$), one may assume that $p_{Y_T} \approx p_{\text{prior}}$ for sufficiently large $T$. Note, however, that unlike in classical diffusion models, which assume access to data from $p_{\text{target}}$, we cannot simulate the noising process. Consequently, the process $Y$ should be understood as an auxiliary construction that facilitates the derivation of our method, rather than as a simulatable component of the algorithm.

The corresponding sampling dynamics, which can be simulated, is given by

$$\mathrm{d}X_s^u = \big(f + \sigma u\big)(X_s^u, s)\,\mathrm{d}s + \sigma(s)\,\mathrm{d}W_s, \quad X_0^u \sim p_{Y_T}, \tag{3}$$

where the objective is to learn a function $u$ that reverses the dynamics of (2), ensuring that $X_T^{u^*} \sim p_{\text{target}}$. It is well known that the optimal solution is given by the (scaled) *score function* $u^* =$

$\sigma^\top \nabla \log \overleftarrow{p}_Y$ (Nelson, 1967; Anderson, 1982). Moreover, as shown in Berner et al. (2024), this function can be equivalently characterized via the following PDE.

**Lemma 2.1** (Hamilton–Jacobi–Bellman PDE for the log-density). *Let $V := -\log \overleftarrow{p}_Y$. Then*

$$\partial_t V + \tfrac{1}{2}\operatorname{Tr}(\sigma\sigma^\top \nabla^2 V) + f \cdot \nabla V - \operatorname{div}(f) - \tfrac{1}{2}\left\|\sigma^\top \nabla V\right\|^2 = 0, \quad V(\cdot, T) = -\log p_{\text{target}}. \quad (4)$$

It is important to note that the boundary term may be replaced by its unnormalized version $-\log \rho_{\text{target}}$. This substitution merely shifts the PDE solution by an additive constant, which is irrelevant for the optimal control since only the gradient of the solution enters.

While equation (4) formally provides a recipe for approximating the log-density and, consequently, the score function, it is well known that solving PDEs in high dimensions is notoriously challenging. In the following, we therefore introduce a procedure that combines Monte Carlo estimation with suitable function approximation to efficiently handle high-dimensional settings.

### 2.2 BSDE REPRESENTATIONS AND BACKWARD ITERATIONS

One approach to approximate certain nonlinear high-dimensional PDEs relies on backward stochastic differential equations (Pardoux, 1998). Their main idea relies on Itô's formula, which, for a stochastic process $X^u$ as defined in (3), states that

$$V(X_T^u, T) - V(X_0^u, 0)$$
$$= \int_0^T \left(\partial_t V + \tfrac{1}{2}\operatorname{Tr}(\sigma\sigma^\top \nabla^2 V + (f + \sigma u) \cdot \nabla V)(X_s^u, s)\right) \mathrm{d}s + \int_0^T \sigma^\top \nabla V(X_s^u, s) \cdot \mathrm{d}W_s. \quad (5)$$

Now, assuming that $V$ solves the HJB PDE (4), we may equivalently write

$$\operatorname{BSDE}(V) := V(X_0^u, 0) - V(X_T^u, T) + \int_0^T h(u, \nabla V)(X_s^u, s) \,\mathrm{d}s + \int_0^T \sigma^\top \nabla V(X_s^u, s) \cdot \mathrm{d}W_s = 0,$$

where for notational convenience we defined the nonlinear term

$$h(u, \nabla V) := \operatorname{div}(f) + \tfrac{1}{2}\left\|\sigma^\top \nabla V\right\|^2 + u \cdot \sigma^\top \nabla V. \quad (6)$$

Conversely, when replacing $V$ by an approximation $\widetilde{V}$, $\operatorname{BSDE}(\widetilde{V}) = 0$ can only hold if $V = \widetilde{V}$ almost surely, by uniqueness of the PDE. This motivates to consider loss functionals of the form

$$\mathcal{L}(\widetilde{V}) := \mathbb{E}\left[\left(\operatorname{BSDE}(\widetilde{V})\right)^2\right], \quad (7)$$

where the expectation is over different realizations of the process $X^u$, cf. E et al. (2017). In principle, $V$ can now be learned by minimizing the loss (7) w.r.t. $V$. However, since solving the minimization problem in its entirety is difficult, it is natural to decompose it into a sequence of smaller subproblems. Following the dynamic programming principle from optimal control theory (Fleming & Rishel, 2012), we thus partition the time horizon into disjoint intervals according to the grid $0 = t_0 < t_1 < \cdots < t_N = T$, and aim to compute $V$ on each interval separately. Our envisioned algorithm proceeds backward in time, beginning with the final interval $[t_{N-1}, t_N]$ and employing the terminal condition $V(\cdot, T) = -\log p_{\text{target}}$. Next, for each preceding interval, the terminal condition is given by the solution obtained at the right endpoint of the subsequent interval. In practice, this means that $V(\cdot, t_n)$ is replaced by its previously computed approximation $\widetilde{V}(\cdot, t_n)$ for $n = 1, \ldots, N-1$. For further details we refer to Algorithm 1 and Richter et al. (2021; 2024).

In practice, we need to discretize both the process (3) and the (stochastic) integrals appearing in the loss (7). Using the same time grid as before, we approximate the solution only at the grid points, i.e., we seek functions $\widehat{V}_n$ such that $\widehat{V}_n \approx V(\cdot, t_n)$ for $n = 0, \ldots, N-1$. To this end, we employ the Euler-Maruyama scheme

$$\widehat{X}_{n+1}^u = \widehat{X}_n^u + \left(f + \sigma u\right)(\widehat{X}_n^u, t_n)\,\Delta t + \sigma(t_n)\,\xi_{n+1}\sqrt{\Delta t}, \quad (8)$$

where $\Delta t = t_{n+1} - t_n$ is the time step and $\xi_{n+1} \sim \mathcal{N}(0, \operatorname{Id})$ are independent standard normal random variables.

We then define for each $n = 0, \ldots, N-1$ the discrete loss

$$\widehat{\mathcal{L}}_n(\widehat{V}_n) = \mathbb{E}\left[\left(\widehat{V}_n(\widehat{X}_n^u) + (\sigma^\top(t_n)\nabla\widehat{V}_n(\widehat{X}_n^u)) \cdot \xi_{n+1}\sqrt{\Delta t} + h_{n+1}^u \Delta t - \widehat{V}_{n+1}(\widehat{X}_{n+1}^u)\right)^2\right], \quad (9)$$

---

**Algorithm 1** Approximation of HJB PDE and optimal control

---

**Input:** Initial parametric choice for the functions $\widehat{V}_n^{(0)}$ and educated guess for $\widehat{u}_n^{(0)}$ for $n \in \{0, \ldots, N-1\}$. Number of outer iterations $I$.
**Output:** Approximation of $V(\cdot, t_n) \approx \widehat{V}_n^{(I)}$ and $u^*(\cdot, t_n) \approx \widehat{u}_n^{(I)}$ for $n \in \{0, \ldots, N-1\}$.
**for** $i = 1$ **to** $I$ **do**
    Simulate $K$ samples of the discretized SDE $\widehat{X}^{u^{(i-1)}}$ according to (8).
    Choose $\widehat{V}_N^{(i)} = -\log \rho_{\text{target}}$.
    **for** $n = N - 1$ **to** $0$ **do**
        Approximate $\widehat{\mathcal{L}}_n(\widehat{V}_n^{(i)})$ as in (9) using Monte Carlo.
        Minimize this quantity (explicitly or by iterative schemes).
        Set $\widehat{V}_n^{(i)}$ to be the minimizer, set $\widehat{u}_n^{(i)} := -\sigma^\top(t_n)\nabla\widehat{V}_n^{(i)}$.
    **end for**
**end for**

---

where, in analogy to (6), $h_{n+1}^u := \left(\text{div}(f) + \frac{1}{2}\|\sigma^\top\nabla\widehat{V}_{n+1}\|^2 + u \cdot \sigma^\top\nabla\widehat{V}_{n+1}\right)(\widehat{X}_{n+1}^u, t_{n+1})$ is introduced for notational convenience. The loss (9) is obtained by discretizing its continuous version (7) by choosing the right endpoint of the deterministic integral and the left endpoint of the stochastic integral. As a next step, the expectation in (9) is discretized by $K \in \mathbb{N}$ sample sequences $(\widehat{X}_n^{u,(k)})_n$ for $k = 1, \ldots, K$, generated from (8). Defining $\Sigma_n^{(k)} = \sigma(t_n)\xi_{n+1}^{(k)}\sqrt{\Delta t} \in \mathbb{R}^d$ and $y_{n+1}^{(k)} = \widehat{V}_{n+1}(\widehat{X}_{n+1}^{u,(k)}) - h_{n+1}^{u,(k)}\Delta t$, the loss in (9) can be approximated as

$$\widehat{\mathcal{L}}_n(\widehat{V}_n) \approx \widehat{\mathcal{L}}_n^K(\widehat{V}_n) := \frac{1}{K}\sum_{k=1}^K \left(\widehat{V}_n(\widehat{X}_n^{u,(k)}) + \Sigma_n^{(k)} \cdot \nabla\widehat{V}_n(\widehat{X}_n^{u,(k)}) - y_{n+1}^{(k)}\right)^2. \qquad (10)$$

Note the affine dependence in $\widehat{V}_n$, making it well-suited for regression-based methods. Moreover, up to discretization error, the loss has the convenient property that it is almost surely zero at the solution, additionally implying vanishing variance at the optimum. For further analysis we refer to Richter et al. (2024).

**Remark 2.2** (Gradient approximation). A key feature of the loss (9) is its explicit dependence on both the function $\widehat{V}_n$ and its gradient $\nabla\widehat{V}_n$. Because the loss penalizes errors in both terms, minimizing it provides a direct incentive to accurately approximate both the function and its derivative. This is particularly advantageous, as the optimal control relies directly on this gradient.

**Remark 2.3** (Error propagation). When decomposing the problem into subproblems, one must take into account the possible propagation of approximation errors. These arise because, in practice, the approximation $\widehat{V}_{n+1}(\cdot) \approx V(\cdot, t_{n+1})$, which serves as a "terminal condition" for computing the individual losses, is generally not exact (see Section 8.3.3 in Gobet (2016)).

**Remark 2.4** (Solving the PDE along a random grid). Unlike traditional numerical methods for PDEs, BSDE-based approaches can be interpreted as operating on dynamically adaptive random grids, provided by the grid points $(\widehat{X}_n^{u,(k)})_{k,n}$. This perspective highlights their potential to achieve dimension-free Monte Carlo convergence rates.

**Remark 2.5** (Fewer target evaluations). A strategy we adopt later is to first approximate the (unnormalized) log-target $\rho_{\text{target}}$ via a simple regression task. The resulting surrogate model can then be employed both for potential Langevin initialization runs and within the outer loop of Algorithm 1. This approach has the advantage that only a potentially small number of target evaluations are required initially; see also Remark 3.2.

Remark 2.4 highlights an additional challenge in sampling applications: we only need to approximate $V$, and hence the control $u$, accurately along the trajectories of $X^u$. However, during training we learn $X^u$ along one set of sampled trajectories, while the actual evaluation later takes place along potentially different trajectories. To address this mismatch, we propose an iterative learning scheme with an outer loop that begins from an informed initial guess $u^{(0)}$ (typically corresponding to Langevin dynamics) and, at each step, updates the control using the solution $u^{(i)}$ obtained in the $i$-th iteration. This iterative refinement is expected to gradually align the training and evaluation

distributions, so that the trajectories used for learning become sufficiently close to those relevant for sampling. In this way, the procedure concentrates on the regions of the state space that are most important for the optimal sampling process; see Algorithm 1 and Figure 1 for an illustration. We note that in our numerical experiments in Section 4 the algorithm typically converges in less than 3 iterations. Furthermore, we refer to Appendix A.4.4 for strategies to extend the learned control in a principled way beyond the data domain.

## 3 TENSOR TRAINS AS APPROXIMATING FUNCTIONS

This section is devoted to the introduction of low-rank tensor formats for approximating the functions $\widehat{V}_n$ introduced above. In particular, we focus on functional Tensor Trains (FTTs, Oseledets (2013)). For discussions regarding the approximation of functions with low-rank formats we refer to Ali & Nouy (2020a;b; 2021); Bachmayr et al. (2021); Griebel et al. (2023); Bachmayr (2023). The FTT format is a specific instance of a nonlinear approximation class designed to exploit separability in the coupling of input parameters of the target function. One of the key advantages of this representation stems from its numerical realization via tensor trains, which form a Riemannian manifold and enable efficient and numerically stable algorithms, including optimization (Oseledets, 2011). When the function to be approximated exhibits a separable structure, the FTT format provides a promising framework for constructing accurate and easily evaluable representations. Let $D = \bigtimes_{i=1}^{d}[a_i, b_i] \subset \mathbb{R}^d$ with $a_i < b_i$ for $i = 1, \ldots, d$. A function $f \colon D \to \mathbb{R}$ is said to have FTT rank $\overline{r} = (\overline{r}_1, \ldots, \overline{r}_{d-1}) \in \mathbb{N}^{d-1}$, if it can be written as

$$f(\boldsymbol{x}) = f(x_1, \ldots, x_d) = F_1(x_1)F_2(x_2) \cdots F_d(x_d) \tag{11}$$

with matrix valued functions $F_i(x_i) \in \mathbb{R}^{\overline{r}_{i-1}, \overline{r}_i}$ for $i = 1, \ldots, d$ with the convention $\overline{r}_0 = \overline{r}_d = 1$. Note that in the case $d = 2$, the FTT format (11) can be interpreted as a singular value decomposition (SVD) in function space, since $f(x_1, x_2) = F_1(x_1)F_2(x_2) = \sum_{k=1}^{\overline{r}_1}[F_1(x_1)]_k[F_2(x_2)]_k$. Thus, the FTT format can be viewed as a generalization of the SVD to high-dimensional functions, while retaining important properties such as closedness Holtz et al. (2012b).

In order to make this format accessible for approximation, a discretization in each direction $x_i$ will be introduced. To this end, let $\mathcal{H} = \bigotimes_{i=1}^{d} \mathcal{H}_i(a_i, b_i)$ be a product Hilbert space on $D$, where each $\mathcal{H}_i(a_i, b_i)$ is equipped with a scalar product $(\cdot, \cdot)_{\mathcal{H}_i}$. For $\boldsymbol{m} = (m_1, \ldots, m_d) \in \mathbb{N}^d$, and using the notation in Table 1, we define the discrete set of $\mathcal{H}$-orthonormal tensor basis functions

$$\mathcal{B}_{\boldsymbol{m}} := \left\{ \phi_{\boldsymbol{\alpha}} := \bigotimes_{i=1}^{d} \phi_{\alpha_i}^{i} \;\middle|\; \boldsymbol{\alpha} \in [\boldsymbol{m}], \; (\phi_{\alpha_j}^{i}, \phi_{\alpha_k}^{i})_{\mathcal{H}_i} = \delta_{jk}, \quad \begin{matrix} i = 1, \ldots, d \\ j, k = 1, \ldots, m_i \end{matrix} \right\}, \tag{12}$$

with univariate basis functions $\phi_{\alpha_i}^{i} \colon [a_i, b_i] \to \mathbb{R}$. For $f$ with FTT rank $\overline{r}$, we may then approximate

$$f(\boldsymbol{x}) \approx \boldsymbol{C}[\Phi(\boldsymbol{x})] := \sum_{\boldsymbol{\alpha} \in [\boldsymbol{m}]} \boldsymbol{C}[\boldsymbol{\alpha}] \phi_{\boldsymbol{\alpha}}(x_1, \ldots, x_d), \tag{13}$$

for $\Phi(\boldsymbol{x}) = (\phi_j^1(x_1))_{j=1}^{m_1} \otimes \cdots \otimes (\phi_j^d(x_d))_{j=1}^{m_d} \in \mathbb{R}^{\boldsymbol{m}}$ and a tensor array $\boldsymbol{C} \in \mathbb{R}^{\boldsymbol{m}}$ with (algebraic) Tensor Train (TT) rank $\boldsymbol{r} = (r_1, \ldots, r_{d-1})^\top \in \mathbb{N}^{d-1}$ bounded by the FTT rank $\overline{r}$. In particular, we have the decomposition into a Tensor Train (or Matrix Product State) format as

$$\boldsymbol{C}[\boldsymbol{\alpha}] = \boldsymbol{C}_1[\alpha_1]\boldsymbol{C}_2[\alpha_2] \cdots \boldsymbol{C}_d[\alpha_d], \tag{14}$$

with order three tensors $\boldsymbol{C}_i \in \mathbb{R}^{r_{i-1}, m_i, r_i}$ defining matrices $\boldsymbol{C}_i[\alpha_i] = \boldsymbol{C}_i[:, \alpha_i, :] \in \mathbb{R}^{r_{i-1}, r_i}$ with the convention that $r_0 = r_d = 1$. The set of TTs of fixed rank $\boldsymbol{r}$ is denoted as $\mathcal{M}_{\boldsymbol{r}}^{\boldsymbol{m}}$. In particular, there exists a maximal possible TT rank $\overline{r} \in \mathbb{N}^{d-1}$ such that $\mathcal{M}_{\overline{r}}^{\boldsymbol{m}} = \mathbb{R}^{\boldsymbol{m}}$.

In what follows, we refer to a function in the form of the right hand side (13) as *extended Tensor Train* (xTT). The xTT format is a special case of the FTT format with a specific choice of univariate basis functions. The set of xTTs building on $\mathcal{M}_{\boldsymbol{r}}^{\boldsymbol{m}}$ is denoted as

$$\mathcal{X}_{\boldsymbol{r}}^{\boldsymbol{m}} := \left\{ \widehat{V}_C = \boldsymbol{C}[\Phi(\cdot)] \mid \boldsymbol{C} \in \mathcal{M}_{\boldsymbol{r}}^{\boldsymbol{m}} \subset \mathbb{R}^{\boldsymbol{m}} \right\}. \tag{15}$$

Note that, by construction, we have $\mathcal{X}_{\boldsymbol{r}}^{\boldsymbol{m}} \subset \mathcal{H}$ for any rank $\boldsymbol{r} \leq \overline{r}$, and for any $\widehat{V}_C \in \mathcal{X}_{\boldsymbol{r}}^{\boldsymbol{m}}$ it holds that $\|\widehat{V}_C\|_{\mathcal{H}} = \|\boldsymbol{C}\|_{\mathrm{F}}$. Key properties of the considered nonlinear low-rank model class

include storage complexity, multi-linearity and related efficient evaluation. Provided that the ranks can be bounded, the TT format exhibits a storage complexity based on the shape of each component tensor $C_i$ with upper bound $\mathcal{O}(\max(m_1, \ldots, m_d)d \max(r_1, \ldots, r_{d-1})^2)$, which scales only linearly in the dimension $d$. Moreover, a function in xTT format can be evaluated fast and numerically stably, including possible access to gradients or Hessians, see Appendix A.4 for details. To this end, originated by (10), we are interested in solving a sequence of optimization tasks for $n = 1, \ldots, N-1$ given by

$$
\min_{\widehat{V}_{\boldsymbol{C}} \in \mathcal{X}_{\boldsymbol{r}}^{\boldsymbol{m}}} \left\{ \widehat{\mathcal{L}}_n^K(\widehat{V}_{\boldsymbol{C}}) + \tau_n \|\widehat{V}_{\boldsymbol{C}}\|_{\mathcal{H}}^2 \right\}
$$

$$
= \min_{\boldsymbol{C} \in \mathcal{M}_{\boldsymbol{r}}^{\boldsymbol{m}}} \left\{ \frac{1}{K} \sum_{k=1}^K \left( \boldsymbol{C}[\Phi(\widehat{X}_n^u)] + \Sigma_n^{(k)} \cdot \nabla \boldsymbol{C}[\Phi(\widehat{X}_n^u)] - y_{n+1}^{(k)} \right)^2 + \tau_n \|\boldsymbol{C}\|_{\mathrm{F}}^2 \right\}
\tag{16}
$$

for some regularization magnitude $\tau_n > 0$. Note that the regularization term involves the $\mathcal{H}$ norm, see Appendix A.3.1. Recall that we are interested in approximating the true value function $V(x, t)$, which is expected to have varying $\mathcal{H}$ norm over time snapshots $t = t_0, \ldots, t_N$. Consequently, to relate the regularized loss from (16) to the original loss from (10), the regularization magnitude needs to be chosen suitably and adaptively, which we explain in detail in Appendix A.6.1. Further, it is crucial to choose the TT rank $\boldsymbol{r}$ and the numbers of degrees of freedom $\boldsymbol{m}$ appropriately, and we provide a corresponding adaptive strategy in Appendix A.6.2 and Appendix A.6.3. Finally, a detailed discussion of choices for tensor basis functions, along with their associated advantages and challenges, is given in Appendix A.3.

### 3.1 Optimization via Alternating Least Squares

The set $\mathcal{M}_{\boldsymbol{r}}$ defines a Riemannian manifold (Holtz et al., 2012b), rendering the option for Riemannian optimization. Here, we consider an alternative approach for minimization based on alternating least squares (ALS), cf. Holtz et al. (2012a). In particular, instead of solving (16) via iterates of the whole TT $\boldsymbol{C}$, we split the minimization into iterations of so-called *sweeps* consisting of several *micro steps*. In each micro step we update the $i$-th component of the TT only, while fixing all other components. For this, we compactly write (see Appendix A.1)

$$
\boldsymbol{C} = \boldsymbol{C}_1 \boldsymbol{C}_2 \cdots \boldsymbol{C}_d = \boldsymbol{U}_1 \cdots \boldsymbol{U}_{i-1} \mathfrak{C} \boldsymbol{U}_{i+1} \cdots \boldsymbol{U}_d
$$

with orthogonal order three tensors $U_j \in \mathbb{R}^{r_{j-1}, m_j, r_j}$ for $j = 1, \ldots, i-1, i+1, \ldots d$ and a so-called *core* $\mathfrak{C} \in \mathbb{R}^{r_{i-1}, m_i, r_i}$, see Appendix A.2 for more details. Then, at each micro step, the loss from (16) reduces to the problem

$$
\min_{\mathfrak{C}} \|A_{i,n}^u \operatorname{vec}(\mathfrak{C}) - y_{n+1}\|_2^2 + \tau_n \|\mathfrak{C}\|_F^2,
\tag{17}
$$

with a matrix $A_{i,n}^u \in \mathbb{R}^{K, r_{i-1} m_i r_i}$ and $y_{n+1} = (y_{n+1}^{(k)})_k \in \mathbb{R}^K$, see Appendix A.5 for a derivation.

After solving the minimization task stated in (16), which yields a TT $\boldsymbol{C}^n \in \mathcal{M}_{\boldsymbol{r}}^{\boldsymbol{m}}$, we then obtain an approximation of our value function $V(\cdot, t_n)$ at time $t = t_n$, denoted by $\widehat{V}_{\boldsymbol{C}^n} \in \mathcal{X}_{\boldsymbol{r}}^{\boldsymbol{m}}$, in xTT format, i.e.

$$
V(\boldsymbol{x}, t_n) \approx \widehat{V}_n(\boldsymbol{x}) \approx \widehat{V}_{\boldsymbol{C}^n} = \boldsymbol{C}^n[\Phi(\boldsymbol{x})], \quad \forall \boldsymbol{x} \in D.
\tag{18}
$$

**Remark 3.1** (Extension strategies for the domain). The approximation of the value function $V$ is only defined locally on a domain $D \Subset \mathbb{R}^d$. In practice, this domain is given by $D = D_n^u$, which depends on both the time step and the current policy $u$. If, during the sampling dynamics in (3), the required policy evaluation $u^* = -\sigma^\top \nabla V$ involves points outside of $D$, suitable extension strategies must be employed. We discuss these strategies in detail in Appendix A.4.4.

**Remark 3.2** (Initialization of xTTs). The subsequent minimization procedure requires initial realizations of the xTTs. For the first time step $t = t_N$, we employ a classical $L^2$ regression of the log-target density using the final samples obtained from the initial sampling dynamics. Assuming that an approximate xTT at time step $t_{n+1}$ is already available, the initial candidate at time $t_n$ is constructed by performing a generalized Galerkin-type projection of the optimized xTT at $t_{n+1}$, while accounting for the change of domain. Further details are provided in Appendix A.4.5.

**Remark 3.3** (Stabilization of backward regressions). While our backward iteration in Algorithm 1 in combination with FTTs offers clear advantages over iterative loss minimization with neural networks, it also introduces additional challenges that can affect the stability of the optimization. Identifying effective strategies to address these issues constitutes an essential contribution of our work, and

we describe these strategies in detail in the appendix. As discussed before, key aspects include the choice of suitable basis functions (Appendix A.3), handling moving domains along sample trajectories and dealing with evaluations outside these domains (Appendices A.4.4 and A.4.5), an adaptive regularization scheme (Appendix A.6.1), adaptive rank selection (Appendix A.6.2) and adaptive basis degree selection (Appendix A.6.3).

# 4 NUMERICAL EXPERIMENTS

In this section we evaluate our sampling algorithm on challenging high-dimensional problems. We refer to an additional problem from computational statistics in Appendix B.5.

## 4.1 HIGHDIMENSIONAL MULTIWELL PROBLEMS

We first consider a class of Multiwell problems, defined by

$$\rho_{\text{target}}(x) = \exp\left(-\sum_{i=1}^{m}(x_i^2 - \delta)^2 - \frac{1}{2}\sum_{i=m+1}^{d} x_i^2\right), \tag{19}$$

which are commonly used benchmarks (Berner et al., 2024; Richter & Berner, 2024; Wu et al., 2020; Midgley et al., 2022). These densities are chosen because they resemble physics and molecular dynamics problems, providing an indication of performance in practical applications. At the same time, they offer a controlled setting for systematically studying the three key challenges in sampling: (1) the dimensionality $d$, (2) the mode separation $\delta$, and (3) the number of modes $2^m$. In our experiments, we use a Fourier basis (see Appendix A.3.1) with 3 to 13 basis functions per dimension, treating this as a hyperparameter tuned according to the observed approximation quality. We employ an $H^2_{\text{mix}}$ orthonormalization as described in Appendix A.3.2. For the noising process (2), we set $f(x,t) = x$, $\sigma(t) = \sqrt{2}$, and consider a time horizon $T = 2$. We apply Algorithm 1 to two Multiwell problems, with dimensions $d = 10$ and $d = 50$ and 9 and 32 modes, respectively, with $\delta = 2$.

As shown in Figure 2, increasing the number of steps $N$ (which also determines the number of functions $\widehat{V}_n$ approximated) steadily improves sampling performance, ultimately achieving remarkable quality. Importantly, the algorithm's runtime is significantly faster than that of alternative, predominantly neural network-based samplers. In $d = 10$, training ranges roughly from 1 minute ($N = 2^8$) to 20 minutes ($N = 2^{13}$), while in $d = 50$ it ranges from 10 to 300 minutes. Note that the runtime grows linearly with $N$. We refer to Figure 3 for a comparison with the *time-reversed diffusion sampler* (DIS) from Berner et al. (2024), which considers the same noising SDE, however relies on neural network approximations and variational training. We refer to Figure 14 in Appendix B.4 for a comparison to further sampling methods, showing significantly improved performance. Finally, we refer to Figure 4 comparing marginals of the 50 dimensional problem with a reference solution.

**Remark 4.1** (Approximation of $\log Z$ and the PDE solution)**.** We note that the Girsanov theorem relates the error in the log-normalizing constant $\log \mathcal{Z}$ to the $L^2$-approximation of the ground-truth score $\nabla \log p$, i.e., the gradient of the solution to the HJB equation w.r.t. to the measure induced by $p_{\text{target}}$ (Berner et al., 2024). Under suitable assumptions, one can further leverage Poincaré inequalities to relate this error to the $L^2$-approximation of the PDE solution itself, measuring the accuracy of our proposed PDE solver.

## 4.2 $\phi^4$ SCALAR FIELD THEORY

Next, we evaluate our method on the discretized Ginzburg-Landau model (specifically, the $\phi^4$ scalar field theory), a canonical framework in statistical mechanics used to describe continuous phase transitions and spontaneous symmetry breaking (Ginzburg & Landau, 1950; Rosenstein & Li, 2021). The target density is defined by a Gibbs distribution over a lattice, characterized by a highly non-convex energy landscape resulting from the competing forces of local bistability and nearest-neighbor spatial coupling,

$$\rho_{\text{target}}(x) = \exp\left(-\frac{1}{2}\sum_{i=1}^{m}(x_i^2 - \delta)^2 - \frac{1}{2}\sum_{i=m+1}^{d} x_i^2 - \frac{1}{2}\sum_{i=1}^{d-1}(x_i - x_{i+1})^2\right). \tag{20}$$

This system presents a formidable challenge for sampling algorithms, as the resulting distribution features exponentially many metastable modes separated by high-energy barriers, typically caus-

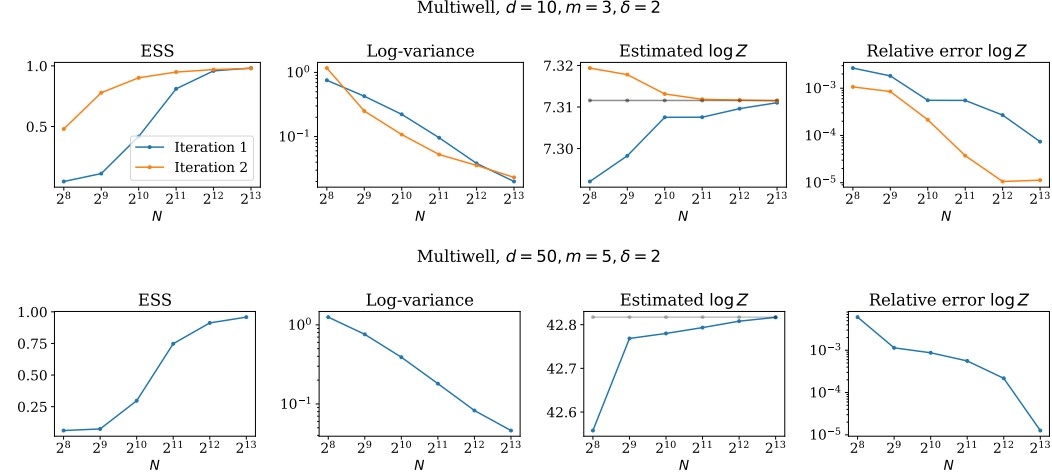

Figure 2: We consider two instances of the Multiwell problem defined in (19) and evaluate performance using the effective sample size (ESS), log-variance divergence, as well as the log-normalizing constant and its relative error; see Appendix B.2. As expected, increasing the number of steps $N$ leads to improved performance and the outer loop can provide additional gains. Remarkably, both the ESS and relative error remain stable even in the more challenging $d = 50$ case.

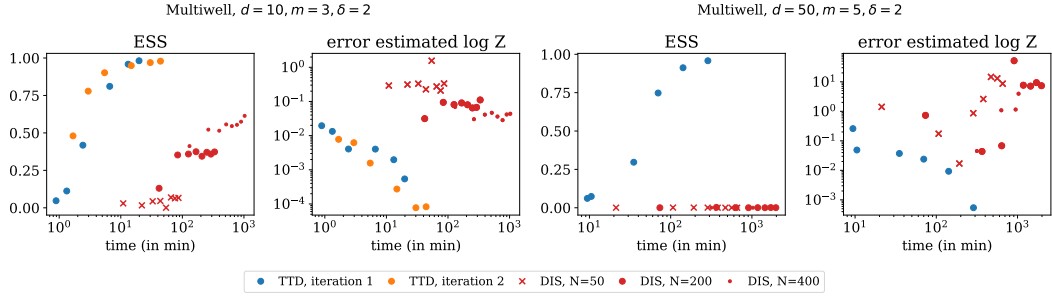

Figure 3: We compare the performance versus runtime of our TTD sampler with DIS (Berner et al., 2024). By design, our algorithm produces one result per chosen number of steps $N$ (shown as blue and orange dots), whereas DIS can improve over training time. Accordingly, we evaluate DIS at equally spaced runtime intervals. In both experiments, our algorithm is not only significantly faster but also achieves better results, particularly in settings where DIS can exhibit instability.

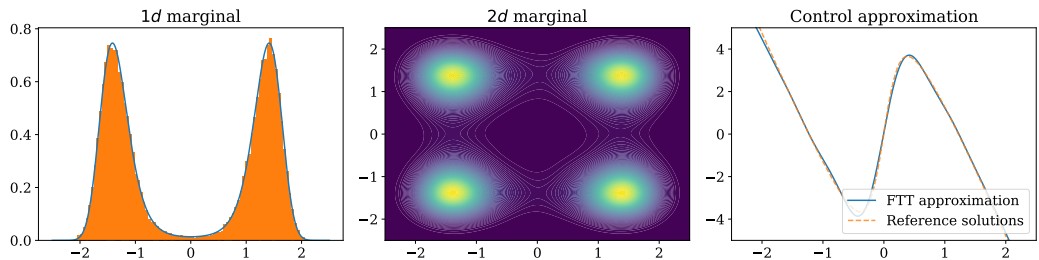

Figure 4: We plot one- and two-dimensional marginals of our Multiwell problem in $d = 50$, showing very high sampling accuracy that aligns with the metrics in Figure 2. Notably, our algorithm does not exhibit mode collapse, a common challenge in diffusion-based sampling (cf. Figure 1). On the right-hand side, we compare the first component of the learned control with a reference solution (computed via finite differences on one-dimensional problems) at $t = 1.9$ for $x$-values varying along a single dimension.

ing standard MCMC chains to suffer from severe mode collapse. Furthermore, it exhibits nearest-neighbor interactions, which create strong dependencies between lattice sites, meaning the sampler

must now contend with both the high-energy barriers of the double wells and a highly constrained, correlated geometry in the high-dimensional state space. Setting $\delta = 2$, we observe in Figure 5 that our TTD method produces high-quality samples from the target distribution in two distinct multi-modal settings.

Ginzburg-Landau model

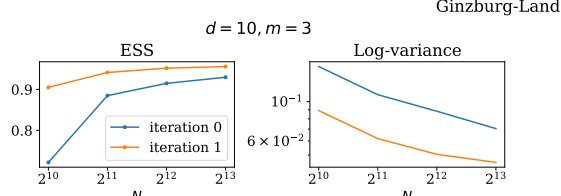
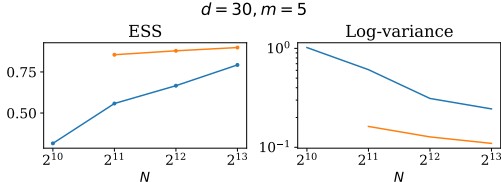

Figure 5: We consider two instances of the Ginzburg–Landau model defined in (20) and assess performance using the effective sample size (ESS) and the log-variance divergence. In both settings, we obtain strong results, with performance improving as the number of steps $N$ and the number of outer iterations increase.

### 4.3 INVESTIGATION OF OPTIMAL RANKS

In this section, we examine highly anisotropic Gaussian target distributions to study the rank structure of the resulting transport problems and to assess the adaptivity of our algorithm. Consider a target density potential of the form $\log \rho_{\text{target}}(x) = -x^\top M x$, where $M$ is a random full rank matrix. For this class of targets, the maximal admissible rank is $\mathbf{r} = (3, 4, \ldots, 2 + d/2, \ldots, 4, 3) \in \mathbb{N}^{d-1}$ for even $d$, see Corollary A.6 for the general case. Since the target distribution evolves toward the standard normal, i.e., $p_{Y_T} \approx p_{\text{prior}} = \mathcal{N}(0, \text{Id})$, the ranks are expected to converge to the constant vector $\mathbf{r} = (2, \ldots, 2) \in \mathbb{N}^{d-1}$. This behavior is confirmed in Figure 6, where we show the evolution of the singular values and the associated rank-adaptivity, ultimately leading to a runtime reduction of our algorithm. The pronounced spectral gaps illustrate that our algorithm successfully identifies and adapts to the appropriate ranks over time, as further visualized in Figure 11 in the appendix.

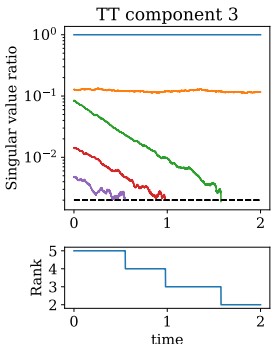

Figure 6: We display the singular values of TT component 3 over time and reduce the rank whenever a corresponding value is below a prespecified threshold.

## 5 CONCLUSION & DISCUSSION

In this work, we introduced Tensor Train Diffusion (TTD), an efficient PDE solver for high-dimensional HJB equations leading to a novel way of sampling from unnormalized densities. Our method solves the HJB equation underlying the diffusion process using a functional tensor train (FTT) representation. By integrating this with a backward-in-time iterative scheme derived from BSDEs, TTD provides a fast, accurate, and stable alternative to neural network techniques. In particular, TTD requires less target evaluations and does not rely on SGD-based optimization with long training times and hyperparameter sensitivity.

In general, our TTD framework presents a solid foundation for the robust and efficient solution of HJB equations, that can readily be used for different sampling problems – we refer to potential limitations in Appendix B.1. While we provide a significantly accelerated tensor train implementation in PyTorch, we believe that further hardware-specific optimization is possible. On the theoretical side, our work motivates further research to analyze latent low-rank structures of different target densities for educated choices of basis functions and variable orderings. Future work could also combine the framework with neural network models or classical sampling methods, as well as extend TTD to broader classes of PDEs and applications. From a theoretical perspective, an open question concerns the behavior of FTT ranks for solutions of the underlying HJB equation, which is only partially understood (Gruhlke et al., 2025). In the general case, the current state-of-the-art theory (Ali & Nouy, 2020a;b; 2021; Griebel et al., 2023; Bachmayr, 2023) does not directly apply and would need to be extended to weighted regularity classes, since solutions of HJB equations are defined on $\mathbb{R}^d$ and are generally unbounded as $\|x\| \to \infty$. Consequently, on the computational side, one must rely on rank-adaptive approaches, i.e., determining the xTT ranks for the approximation in an adaptive manner.

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

APPENDIX CONTENTS

## A   EXTENDED TENSOR TRAINS

This section will give a deeper understanding in the concepts of tensor trains and extended tensor trains. We note that the concept of an extended tensor train in dimension $d = 1$ reduces to the approximation a linear vector space of basis functions. For $d = 2$ the concept of tensor trains coincides with singular value decomposition. Since singular value decomposition in higher dimension, i.e. the decomposition of a tensor into sums of rank-1 tensors is not a closed format, suitable for well-defined optimization tasks, we utilize the framework of tensor trains, which form a closed Riemannian manifold in $\mathbb{R}^{\boldsymbol{m}}$ (Holtz et al., 2012b).

### A.1   CONTRACTION OF TENSORS

Let $\boldsymbol{m} \in \mathbb{N}^d$ be the *mode sizes*. For a tensor $\boldsymbol{A} \in \mathbb{R}^{\boldsymbol{m}}$, where refer to the tensor slice in the $k$-th dimension as the $k$-th mode dimension. A tensor with only one mode dimension is a vector, a tensor with only two mode dimensions is a matrix, a tensor with three mode dimensions is a order three tensor, etc.

| | |
|---|---|
| $\boldsymbol{m} \in \mathbb{N}^d$ | dimension array $\boldsymbol{m} = (m_1, \ldots, m_d)$ |
| $k\boldsymbol{m} + l$ | $(km_1 + l, \ldots, km_d + l)$ for $k, l \in \mathbb{N}_0$ |
| $[\boldsymbol{m}]$ | indexing $[\boldsymbol{m}] = \times_{i=1}^{d}\{1, \ldots, m_i\}$ |
| $\boldsymbol{m}_1 \geq \boldsymbol{m}_2, \boldsymbol{m} \geq k$ | component wise comparison $\boldsymbol{m}, \boldsymbol{m}_1, \boldsymbol{m}_2 \in \mathbb{N}^d, k \in \mathbb{N}$ |
| $\boldsymbol{\alpha}, \boldsymbol{\beta}, \boldsymbol{\gamma}$ | multiindex in $\mathbb{N}_0^d$, note that we always index starting from 0 |
| $\mathbb{R}^{\boldsymbol{m}}$ | tensor space $\mathbb{R}^{m_1, \ldots, m_d}$ |
| $\|\cdot\|_{\mathrm{F}}$ | Frobenius norm |
| $\boldsymbol{A}, \boldsymbol{B}, \boldsymbol{C}$ | tensor elements in $\mathbb{R}^{\boldsymbol{m}}$ |
| $\boldsymbol{r}$ | rank $\boldsymbol{r} = (r_1, \ldots, r_{d-1})$ in $\mathbb{N}^{d-1}$ |
| $\boldsymbol{r}^1 \boldsymbol{r}^2$ | multiplication $\boldsymbol{r}^1 \boldsymbol{r}^2 = (r_1^1 r_1^2, \ldots, r_{d-1}^1 r_{d-1}^2)$ in $\mathbb{N}^{d-1}$ |
| $k_i, l_i$ | rank enumeration indices in $\{1, \ldots, r_i\}$ |
| $A_i, B_i, C_i$ | component order 3 tensor in $\mathbb{R}^{r_{i-1}, m_i+1, r_i}$ with entries indexed by $[k_{i-1}, \alpha_i, k_i]$ |
| $A_i[\alpha_i]$ | matrix extraction $A_i[\alpha_i] = A_i[:, \alpha_i, :] \in \mathbb{R}^{r_{i-1}, r_i}$ of component tensor $A_i$ |
| $A_i[k_{i-1}, :, k_i]$ | vector extraction in $\mathbb{R}^{m_i+1}$ for each rank enumeration $k_{i-1}, k_i$ |
| $\boldsymbol{A}[\boldsymbol{\alpha}]$ | tensor indexing $\boldsymbol{A}[\alpha_1, \ldots, \alpha_d]$ for $\boldsymbol{A} \in \mathbb{R}^{\boldsymbol{m}}, \boldsymbol{\alpha} \in [\boldsymbol{m}], \boldsymbol{m} \in \mathbb{N}^d$ |

Table 1: List of compact notations used in this work.

As first we will define the key concept of *contraction* between order tensors. Let $k, \ell \in \mathbb{N}$ with $\ell > k$. For any tensor $\boldsymbol{A} \in \mathbb{R}^{m_1, \ldots, m_k}$ and $\boldsymbol{B} \in \mathbb{R}^{m_k, \ldots, m_\ell}$, we simply define $\boldsymbol{AB} \in \mathbb{R}^{m_1, \ldots, m_{k-1}, m_{k+1}, \ldots, m_\ell}$ as the tensor resulting from the contracting of neighboured mode dimensions, i.e. the last mode dimension of $\boldsymbol{A}$ and the first of $\boldsymbol{B}$:

$$\boldsymbol{AB} = \sum_{i_k=1}^{m_k} \boldsymbol{A}[\ldots, i_k]\boldsymbol{B}[i_k, \ldots],$$

where we used Python type notation for easier understanding.

More general for two tensors $\boldsymbol{C} \in \mathbb{R}^{\boldsymbol{m}^1}$ and $\boldsymbol{D} \in \mathbb{R}^{\boldsymbol{m}^2}$ for $\boldsymbol{m}^1 \in \mathbb{N}^{d_1}$ with $\boldsymbol{m}^1 = (m_1^1, \ldots, m_{d_1}^1)$ and $\boldsymbol{m}^2 \in \mathbb{N}^{d_2}$ with $\boldsymbol{m} = (m_1^2, \ldots, m_{d_2}^2)$ such that the $k$-th and the $\ell$-th mode size $m_k^1$ and $m_\ell^2$ coincide, i.e. $m = m_k^1 = m_\ell^2$, we define the contraction with respect to the $k$-th and $\ell$-th mode direction as the tensor $\boldsymbol{E} \in \mathbb{R}^{m_1^1, \ldots, m_{k-1}^1, m_{k+1}^1 m_{d_1}^1, m_1^2, \ldots, m_{\ell-1}^2, m_{\ell+1}^2, m_{d_2}^2}$ with

$$\boldsymbol{E} = \boldsymbol{C} \circ_{kl} \boldsymbol{D} = \sum_{i=1}^{m} \underbrace{\boldsymbol{C}[\ldots, i, \ldots]}_{k\text{-th slice}} \underbrace{\boldsymbol{D}[\ldots, i, \ldots]}_{\ell\text{-th slice}} \in .$$

In particular, using this notation it holds that

$$\boldsymbol{AB} = \boldsymbol{A} \circ_{k,1} \boldsymbol{B}.$$

For further readings, we refer to a generalized framework and unified notation to contractions in general tensor networks, which can be found in Gruhlke & Moser (2025).

### A.2 DEGREES OF FREEDOM, THE CORE AND REMAINING ORTHOGONAL COMPONENTS

This section is devoted to representation of Tensor Trains (TTs) suitable for numerically stable usage, in particular for the high dimensional case.

Consider a TT with TT rank $\boldsymbol{r} = (r_1, \ldots, r_{d-1}) \in \mathbb{N}^{d-1}$ given as

$$\boldsymbol{C}[\boldsymbol{\alpha}] = \boldsymbol{C}_1[\alpha_1]\boldsymbol{C}_2[\alpha_2] \cdots \boldsymbol{C}_d[\alpha_d], \forall \boldsymbol{\alpha} \in [\boldsymbol{m}] \tag{21}$$

with $\boldsymbol{C}_i \in \mathbb{R}^{r_{i-1}, m_i, r_i}$ with the convention of $r_0 = r_d = 1$.

Using the notation of contractions of tensors from Appendix A.1, we can compactly write

$$\boldsymbol{C} = \boldsymbol{C}_1 \boldsymbol{C}_2 \cdots \boldsymbol{C}_d \tag{22}$$

First, we note that for any $i = 1, \ldots, d-1$ you can choose an arbitrary invertible matrix $G_i \in \mathrm{GL}(r_i) \subset \mathbb{R}^{r_i, r_i}$ and insert it and its inverse between the $i$-th and $i+1$th component without changing the represented full tensor. In particular, let

$$\widehat{\boldsymbol{C}}_i = \boldsymbol{C}_i G_i, \qquad \widehat{\boldsymbol{C}}_{i+1} = G_i^{-1} \boldsymbol{C}_{i+1}.$$

Then,

$$\boldsymbol{C} = \boldsymbol{C}_1 \boldsymbol{C}_2 \cdots \boldsymbol{C}_d = \boldsymbol{C}_1 \cdots \widehat{\boldsymbol{C}}_i \widehat{\boldsymbol{C}}_{i+1} \cdots \boldsymbol{C}_d. \tag{23}$$

The space $\mathrm{GL}(r_i)$ is of dimension $r_i^2$ and consequently, the degrees of freedoms in a TT is given as

$$\#\mathrm{d.o.f.(TT)} = \underbrace{\sum_{i=1}^{d} r_{i-1} m_i r_i}_{\text{raw number of entries for each components}} - \underbrace{\sum_{i=1}^{d-1} r_i^2}_{\text{dimensions of } \mathrm{GL}(r_i)} \tag{24}$$

For further readings to the derivation of degrees of freedoms in terms of gauge conditions, we refer to Oseledets (2011); Holtz et al. (2012b); Uschmajew & Vandereycken (2013).

Now, we can apply high-order singular value decomposition (HOSVD, see Grasedyck (2010); Oseledets (2011)) on $\boldsymbol{C}$. Let us fix $1 \leq k \leq d$ as the so-called *core position*. Then for any $i = 1, \ldots, k-1$, iteratively we reshape the $i$-th TT component $\boldsymbol{C}_i \in \mathbb{R}^{r_{i-1}, m_i, r_i}$ as a matrix $C_i \in \mathbb{R}^{r_{i-1}, m_i r_i}$ apply a singular value decomposition

$$C_i = U_i \Sigma_i V_i^\top,$$

and contract the non-orthonormal part $\Sigma_i V_i^\top$ from left to $\boldsymbol{C}_{i+1}$ and redefine

$$\boldsymbol{C}_{i+1} = \Sigma_i V_i^\top \boldsymbol{C}_{i+1}.$$

Analogously, for $i = d, d-1, \ldots, k+1$ we iteratively perform SVDs $C_i = U_i \Sigma_i V_i^\top$ and contract $U_i \Sigma_i$ from right to $\boldsymbol{C}_{i-1}$. Finally we can reshape each $U_i \in \mathbb{R}^{r_{i-1}, m_i r_i}$ to a order three tensor $\boldsymbol{U}_i \in \mathbb{R}^{r_{i-1}, m_i, r_i}$ for $i = 1, \ldots, k-1$ and $V_i^\top \in \mathbb{R}^{r_{i-1}, m_i r_i}$ to order three tensor $\boldsymbol{U}_i \in \mathbb{R}^{r_{i-1}, m_i, r_i}$ for $i = d, d-1, \ldots, k+1$. Then, it holds that

$$\boldsymbol{C} = \boldsymbol{U_1} \cdots \boldsymbol{U}_{k-1} \boldsymbol{C}_k \boldsymbol{U}_{k+1} \cdots \boldsymbol{U}_d. \tag{25}$$

The resulting updated non-orthonormal component $\mathfrak{C} = \boldsymbol{C}_k$ is called the *core* at position $k$ of $\boldsymbol{C}$.

The classical representation of a TT from (21) is prone to rounding errors, when trying to access $\boldsymbol{C}[\boldsymbol{\alpha}]$ with $\boldsymbol{\alpha} = (\alpha_1, \ldots, \alpha_d)$. Instead, one first defines a core representation with core $\mathfrak{C}$, e.g. with a core position $k$. Then, the orthogonal components are contracted first and in the end contracted with the core, leading to a numerically stable result. In particular, a left contraction computationally realized from left left to right given as

$$\boldsymbol{L}_k[\alpha_1, \ldots, \alpha_{k-1}] = \boldsymbol{U_1}[\alpha_1] \cdots \boldsymbol{U}_{k-1}[\alpha_{k-1}], \qquad \boldsymbol{L}_k := 1 \text{ for } k = 1, \tag{26}$$

and a right contraction result $\boldsymbol{R}_k$ computationally realized from right to left contractions is defined as

$$\boldsymbol{R}_k[\alpha_k, \ldots, \alpha_d] = \boldsymbol{U}_{k+1}[\alpha_{k+1}] \cdots \boldsymbol{U}_d[\alpha_d], \qquad \boldsymbol{R}_k = 1 \text{ for } k = d. \tag{27}$$

Finally, the contraction reads

$$\boldsymbol{C} = \boldsymbol{L}_k \mathfrak{C} \boldsymbol{R}_k.$$

### A.3 TENSOR BASIS FUNCTIONS

First note that we require basis functions with local support. This is motivated by two aspects due to the approximation of the value function. First, the value function is (up to constant shift) a negative log-density, with unbounded support on $\mathbb{R}^d$. Such functions in general are in no classical integrable smoothness space such as Sobolev spaces. Second, the support of such densities may have localized main mass, this is the area of interest for the localization. In our proposed algorithm, we approximate the value function on sample trajectories. Hence, in the idealistic setup for a value function $V$ and with density $\propto e^{-V}$, and perfect sample trajectories in the regime of exact reverse sampling dynamics at time $t$ we would approximate $V(\cdot, t)$ e.g. in $L^2(\mathbb{R}^d, e^{-V}(\cdot, t)\mathrm{d}\boldsymbol{x})$ or in weighted Sobolev spaces. Consequently, the localization or truncation approach to compact subdomains $K$ then approximates such weighted spaces.

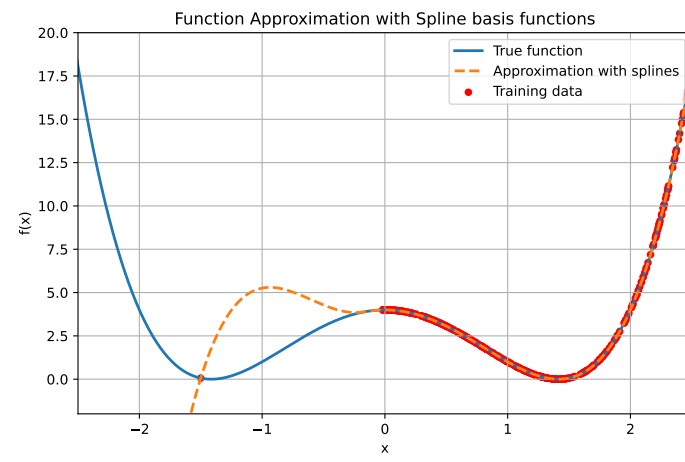

Figure 7: A spline basis with 3 knots, local polynomial degree 4 and smoothness $s = 1$ is used to approximation a target function with limited amount of samples on the left side of origin. Even if there is a soft coupling due to the parameter $s$, the sample regression fails miserably.

### A.3.1 ORTHONORMAL UNIVARIATE BASIS FUNCTIONS

Let $a_i < b_i$ and a Hilbert space $\mathcal{H}_i(a_i, b_i)$ with inner product $(\cdot, \cdot)_{\mathcal{H}_i}$. We will introduce those basis functions and discuss limits and challenges due to approximation, localization power and sample robustness. Then we introduce the concept of orthonormalization. Note that the sample robustness does not depend on the orthonormalization or any other representation of the basis but on their resulting spanned vector space.

**Legendre polynomials.** As a classic basis we consider classical Legendre polynomials $L_0, L_1, L_2, \ldots$.

- *Approximation*: Legendre polynomials admit asymptotic spectral convergence for analytic targets.

- *Localization*: Functions with local features may require high amount of polynomial degree for a proper approximation (the constant dominates the pre-asymptotics). Functions with local features may require high amount of polynomial degree for a proper approximation (the constant dominates the pre-asymptotics).

- *Sample robustness*: For $L^2$ based empirical regression problems, a Legendre basis requires a number of samples polynomial in the degree of freedom to have unique least square solution with high probability, see Cohen & Migliorati (2017). This limits the possible use of a high amount of basis functions due to the amount of required number of samples for regression.

**Spline polynomials of smoothness $s$ and order $p$.** The second basis ansatz involves spline functions of local smoothness $s$ and local polynomial degree $p$ defined on a grid $a_i = x_0 < x_1 < \ldots < x_k = b_i$ with $k+1$ knots implemented via Cox-de Boor recursion formulas, see De Boor & De Boor (1978).

- *Approximation*: Spline functions admit algebraic convergence rates in $p$ based on the grid width Schumaker (2007).

- *Localization*: Localized effects can be captured much better compared to global polynomials.

- *Sample robustness*: As observed in the experiments for empirical regression a well covering ensemble of samples is required for robust regression. In particular, due to the local property of the basis, if a low amount of samples is available between knots, then reconstruction may fail miserably for simple functions, see Appendix A.3.1. Note, that also rank regularization is not enough to correct for the stable regression using splines under various sample distributions. This drawback becomes in particular critical in cases of our algorithm, when the initial policy does not allow for proper mass distribution, e.g. for multimodal targets.

**Fourier modes.** Fourier modes $1, \sin(k\omega x), \cos(k\omega x)$ for $k = 1, 2, \ldots$ are a classic basis class.

- *Approximation*: While Fourier modes admit spectral approximation power for analytic functions, where any derivative has a periodic extension, the approximation power reduces to algebraic rates for non-period functions, e.g. order 1 for smooth non-periodic functions.

- *Localization*: Similar to Legendre polynomials local features may require high amount of Fourier modes for a proper approximation. Non-periodic target functions introduces spurious oscillations in the approximation by Fourier modes.

- *Sample robustness*: Fourier modes are known for its optimal sample complexity and robustness, e.g. compared to Legendre polynomials, see Trunschke (2022). This allows for a high amount of basis functions without explosion in the required number of samples for regression.

**Extended Fourier modes.** This basis involves Fourier modes extended by Legendre polynomials $L_0 \equiv 1, L_1, L_2, \sin(k\omega x), \cos(k\omega x)$ for $k = 1, 2, \ldots$.

- *Approximation*: The enrichment with polynomials up to degree $p$ allows to fix approximation rate drawbacks due to non-periodic issues. For the enrichment of $p = 2$ order Legendre polynomials, the convergence rate for smooth non-periodic functions is algebraic of order $p + 1 = 3$.

- *Localization*: Same as Legendre and Fourier modes.

- *Sample robustness*: We have advantages from both worlds, Legendre and Fourier modes. We expect better sample complexity and robustness than Legendre polynomials, allowing for higher amount of basis functions with controlled need of samples. While a rigorous proof for this is out of the scope of this paper, experimental evidence was made by the authors.

### A.3.2 ORTHONORMAL TENSOR BASIS FUNCTIONS AND CHOICE OF $\mathcal{H}$

For $i = 1, \ldots, d$, let $\Phi_{i,\mathrm{raw}} = (\phi_{\alpha_i,\mathrm{raw}})_{\alpha_i=1}^{m_i} \in \mathbb{R}^{m_i}$ denote the stacked basis vector of $m_i \in \mathbb{N}$ raw univariate basis functions, e.g. Legendre polynomials, Splines, extended Fourier modes. Then we define the associated gramian matrix $G_i \in \mathrm{GL}(m_i)$ with entries defined as

$$[G_i]_{k,\ell} = (\phi_{\alpha_k,\mathrm{raw}}, \phi_{\alpha_\ell,\mathrm{raw}})_{\mathcal{H}_i(a_i,b_i)}, \quad k, \ell = 1, \ldots, m_i.$$

Then

$$(\phi_j^i)_{j=1}^{m_i} := G_i^{-1/2} \Phi_{i,\mathrm{raw}} \tag{28}$$

defines a vector of $\mathcal{H}_i(a_i, b_i)$ orthonormal basis functions.

Then, tensorization of these, yields the basis set

$$\mathcal{B}_{\boldsymbol{m}} := \left\{ \phi_{\boldsymbol{\alpha}} := \bigotimes_{i=1}^d \phi_{\alpha_i}^i \ \middle| \ \boldsymbol{\alpha} \in [\boldsymbol{m}], (\phi_{\alpha_j}^i, \phi_{\alpha_k}^i)_{\mathcal{H}_i} = \delta_{jk}, \forall i, j, k \right\}, \tag{29}$$

which forms a $\mathcal{H}$-orthonormal basis for the product Hilbert space $\mathcal{H} = \bigotimes_{i=1}^d \mathcal{H}_i(a_i, b_i)$.

In this work we utilize $\mathcal{H}_i(a_i, b_i) = H^2(a_i, b_i)$ being the Sobolev space of smoothness 2. It holds for $K = \times [a_i, b_i]$, that

$$\mathcal{H} = H_{\mathrm{mix}}^2(K), \tag{30}$$

where $H_{\mathrm{mix}}^s(K)$ denotes the classical space of mixed regularity of smoothness $s$.

This choice of Hilbert is motivated by two aspects. First the BSDE loss from (9) requires zero -and first order integrable derivatives. Then due to possible oscillatory behavior in the approximation step, we want to regularize the second order derivatives to control oscillations in the first derivative, leading to our approximate policy.

The choice of orthonormalization is motivated to exploit Parseval identity in the optimization scheme, since for $\widehat{V}_C = C[\Phi(\cdot)] \in \mathrm{span}\,\mathcal{B}_{\boldsymbol{m}}$ the identity $\|\widehat{V}_C\|_{\mathcal{H}} = \|C\|_{\mathrm{F}}$ is inherited in the micro steps of the alternating minimization scheme. The latter gives a clear interpretation of the regularization term $\tau_n \|\mathfrak{C}\|_{\mathrm{F}}$ from (17), since for a TT $C$ with core representation as in (25) with core $\mathfrak{C}$ it holds

$$\|C\|_{\mathrm{F}} = \|\mathfrak{C}\|_{\mathrm{F}}. \tag{31}$$

Consequently, $\|\mathfrak{C}\|_{\mathrm{F}} = \|\widehat{V}_C\|_{\mathcal{H}}$ is the $\mathcal{H}$ norm of $\widehat{V}_C$.

### A.4 Evaluation, gradient and Hessian of extended Tensor Trains

This section is devoted to illustrate the efficient and numerically stable evaluation of functions in xTT format involving their gradients and Hessians.

For this let $\widehat{V}_{\boldsymbol{C}} \in \mathcal{X}_{\boldsymbol{r}}^{\boldsymbol{m}}$ be a extended tensor train as defined in (15). Then, it holds using $k_0 = k_d = 1$

$$\widehat{V}_{\boldsymbol{C}}(\boldsymbol{x}) = \boldsymbol{C}[\Phi(\boldsymbol{x})] = \sum_{\boldsymbol{\alpha} \in [\boldsymbol{m}]} \boldsymbol{C}[\boldsymbol{\alpha}] \phi_{\boldsymbol{\alpha}}(x_1, \ldots, x_d)$$

$$= \sum_{k_1=1}^{r_i} \cdots \sum_{k_{d-1}=1}^{r_{d-1}} \boldsymbol{C}_1[k_0, \alpha_1, k_2] \cdots \boldsymbol{C}_1[k_{d-1}, \alpha_1, k_d] \prod_{i=1}^{d} \phi_{\alpha_i}^i(x_i)$$

#### A.4.1 Evaluation

Now define the vector $w^i = w^i(x_i)$ as

$$[w^i(x_i)]_j = ((\phi_j^i(x_i))_{j=1}^{m_i} \in \mathbb{R}^{m_i}.$$

Then,

$$\widehat{V}_{\boldsymbol{C}}(\boldsymbol{x}) = \underbrace{\boldsymbol{C}_1 \circ_{2,1} w^1(x_1)}_{\in \mathbb{R}^{r_0, r_1}} \cdots \underbrace{\boldsymbol{C}_d \circ_{2,1} w^d(x_d)}_{\in \mathbb{R}^{r_{d-1}, r_d}}$$

Now in order to utilize stable evaluations, we consider a core representation as discussed in Appendix A.2.

To that end, we simplify the discussion and assume that the **core position at evaluation** is $k = 0$. Consequently, we have

$$\boldsymbol{C} = \mathfrak{C}\boldsymbol{U}_2 \cdots \boldsymbol{U}_d = \mathfrak{C}\boldsymbol{R}_1. \tag{32}$$

Then, by noting that $r_0 = r_d = 1$ the evaluation is realized as

$$\widehat{V}_{\boldsymbol{C}}(\boldsymbol{x}) = \underbrace{\mathfrak{C} \circ_{2,1} w^1(x_1)}_{\in \mathbb{R}^{1, r_1}} \underbrace{\boldsymbol{U}_2 \circ_{2,1} w^1(x_2)}_{\in \mathbb{R}^{r_1, r_2}} \cdots \underbrace{\boldsymbol{C}_d \circ_{2,1} w^d(x_d)}_{\in \mathbb{R}^{r_{d-1}, 1}}, \tag{33}$$

which contracted from right to left reduces to iterates of matrix-vector multiplications after each $\circ_{2,1}$ contraction is performed first.

#### A.4.2 Gradient evaluation

Additional define the derivative vector $\dot{w}^i = \dot{w}^i(x_i)$ as

$$[\dot{w}^i(x_i)]_j = ((\partial_{x_i}\phi_j^i(x_i))_{j=1}^{m_i} \in \mathbb{R}^{m_i}, \quad i = 1, \ldots, d.$$

Then it holds

$$\partial_{x_i}\widehat{V}_{\boldsymbol{C}}(\boldsymbol{x}) = \boldsymbol{C}_1 \circ_{2,1} w^1(x_1) \cdots \boldsymbol{C}_i \circ_{2,1} \dot{w}^i(x_i) \cdots \boldsymbol{C}_d \circ_{2,1} w^d(x_d)$$

$$= \mathfrak{C} \circ_{2,1} w^1(x_1) \cdots \boldsymbol{U}_i \circ_{2,1} \dot{w}^i(x_i) \cdots \boldsymbol{C}_d \circ_{2,1} w^d(x_d)$$

So a naive approach for gradient evaluations would to perform $d$ of these matrix vector contractions. However, this process can be accelerated avoiding recomputation using intermediate stacks. First motivated by (27), the functional right vector stacks are defined as for $k = d, d-1, \ldots, 1$ as

$$R_k(x_{k+1}, \ldots, x_d) = \boldsymbol{U}_{k+1} \circ_{21} w^{k+1}(x_{k+1}) \cdots \boldsymbol{U}_d \circ_{21} w^d(x_d), \qquad R_k = 1 \text{ for } k = d, \tag{34}$$

where the evaluation is performed from left to right based on matrix vector multiplications. It holds that $R_k(x_{k+1}, \ldots, x_d) \in \mathbb{R}^{r_k, 1}$ is a vector. Note, that the recursive relation holds

$$R_k(x_{k+1}, \ldots, x_d) = \boldsymbol{U}_{k+1} \circ_{21} w^{k+1}(x_{k+1})R_{k+1}(x_{k+2}, \ldots, x_d), \tag{35}$$

avoiding recomputation when building each $R_k$. Moreover for each $\ell = 2, \ldots, d-1$ we defined the intermediate matrix stacks $M_k$ as

$$M_2(x_2) = \boldsymbol{U}_2 \circ_{21} w^2(x_2), \quad M_\ell(x_2, \ldots, x_\ell) = M_{\ell-1}(x_2)\boldsymbol{U}_\ell \circ_{21} w^\ell(x_\ell). \tag{36}$$

Again the recursive relation avoids recomputation. Then, it holds that

$$\partial_{x_1} \widehat{V}_{\boldsymbol{C}}(\boldsymbol{x}) = \mathfrak{C} \circ_{2,1} \dot{w}^1(x_1) R_1(x_2, \dots, x_d), \tag{37}$$

$$\partial_{x_i} \widehat{V}_{\boldsymbol{C}}(\boldsymbol{x}) = \mathfrak{C} \circ_{2,1} w^1(x_1) M_{i-1}(x_2, \dots, x_{i-1}) \boldsymbol{U}_i \circ_{21} \dot{w}^i(x_i) R_i(x_{i+1}, \dots, x_d). \tag{38}$$

Again for numerically stability first $M_{i-1}(x_2, \dots, x_{i-1}) \boldsymbol{U}_i \circ_{21} \dot{w}^i(x_i) R_i(x_{i+1}, \dots, x_d)$ is contracted in (38), refered as the orthogonal block, then a final contraction is realized with the non-orthogonal block involving the core $\mathfrak{C}$.

This formulation improves the gradient evaluation speed up to a factor $d$ compared to naive gradient evaluations.

### A.4.3 HESSIAN EVALUATION

Additional define the second derivative vector $\ddot{w}^i = \ddot{w}^i(x_i)$ as

$$[\ddot{w}^i(x_i)]_j = ((\partial^2_{x_i x_i} \phi^i_j(x_i))_{j=1}^{m_i} \in \mathbb{R}^{m_i}, \quad i = 1, \dots, d.$$

Then it holds

$$\partial^2_{x_i x_i} \widehat{V}_{\boldsymbol{C}}(\boldsymbol{x}) = \boldsymbol{C}_1 \circ_{2,1} w^1(x_1) \cdots \boldsymbol{C}_i \circ_{2,1} \ddot{w}^i(x_i) \cdots \boldsymbol{C}_d \circ_{2,1} w^d(x_d)$$

$$= \mathfrak{C} \circ_{2,1} w^1(x_1) \cdots \boldsymbol{U}_i \circ_{2,1} \ddot{w}^i(x_i) \cdots \boldsymbol{C}_d \circ_{2,1} w^d(x_d)$$

and for $i < j$

$$\partial^2_{x_i x_j} \widehat{V}_{\boldsymbol{C}}(\boldsymbol{x}) = \boldsymbol{C}_1 \circ_{2,1} w^1(x_1) \cdots \boldsymbol{C}_i \circ_{2,1} \dot{w}^i(x_i) \cdots \boldsymbol{C}_j \circ_{2,1} \dot{w}^j(x_j) \boldsymbol{C}_d \circ_{2,1} w^d(x_d)$$

$$= \mathfrak{C} \circ_{2,1} w^1(x_1) \cdots \boldsymbol{U}_i \circ_{2,1} \dot{w}^i(x_i) \cdots \boldsymbol{U}_j \circ_{2,1} \dot{w}^j(x_j) \cdots \boldsymbol{C}_d \circ_{2,1} w^d(x_d).$$

So a naive approach for gradient evaluations would to perform $\mathcal{O}(d^2)$ of these matrix vector contractions. The diagonal Hessian terms evaluation can be accelerated similar to the approach for the gradient evaluation using the right vector stacks $R_k$ and the mid stack $M_{k-1}$. Moreover many sub-contractions reappear for different $i < j$ for the remaining part. Using the right vector stacks $R_k$ we can iteratively build $\dot{R}^j_k$ stacks of the form

$$\dot{R}^j_i(x_{i+1}, \dots, x_d) = U_{i+1} \circ_{2,1} w^{i+1}(x_{i+1}) \dots U_j \circ_{2,1} \dot{w}^j(x_j) R_j(x_{j+1}, \dots, x_d)$$

which again can be build for each $i < j$ recursively including avoiding unnecessary recontractions of $U_{i+1} \circ_{2,1} w^{i+1}(x_{i+1}) \dots U_j \circ_{2,1} \dot{w}^j(x_j)$.

Then the final computation is possible

$$\partial^2_{x_1 x_j} \widehat{V}_{\boldsymbol{C}}(\boldsymbol{x}) = \mathfrak{C} \circ_{2,1} \dot{w}^1(x_1) \dot{R}^j_1(x_2, \dots, x_d), \quad j = 2, \dots, d \tag{39}$$

$$= \mathfrak{C} \circ_{2,1} w^1(x_1) M_{i-1}(x_2, \dots, x_{i-1}) \boldsymbol{U}_i \circ_{2,1} \dot{w}^i(x_i) \dot{R}^j_i(x_{i+1}, \dots, x_d). \tag{40}$$

Again this contraction in (40) is performed from left to right, leading to the contraction of the non-orthonormal portion as the last step.

This formulation improves the Hessian evaluation speed up to a factor $\mathcal{O}(d^2)$ compared to naive Hessian evaluations.

### A.4.4 MOVING APPROXIMATION DOMAIN AND APPROXIMATE POLICY EXTENSIONS

At timestep $t = t_n$, the empirical loss from (10) depends on samples $(\widehat{X}^{u,(k)}_n)^K_{k=1}$. Motivated by the idealistic situation that, up to approximation error,

$$\widehat{X}^{u,(k)}_n \sim e^{-V(\cdot, t_n)},$$

we choose an approximation domain $D_n$ time step dependent, based on the location of samples. To this end, we define

$$\widehat{a}_{i,n} := \min_{k=1,\dots,K} (\widehat{X}^{u,(k)}_n)_i, \qquad \widehat{b}_{i,n} := \max_{k=1,\dots,K} (\widehat{X}^{u,(k)}_n)_i, \qquad i = 1, \dots, d. \tag{41}$$

Then, for a domain extension factor $0 \le q < 1$, we define

$$a_{i,n} := \widehat{a}_{i,n} - q|\widehat{b}_{i,n} - \widehat{a}_{i,n}|, \qquad b_{i,n} := \widehat{b}_{i,n} + q|\widehat{b}_{i,n} - \widehat{a}_{i,n}|, \qquad i = 1, \dots, d. \tag{42}$$

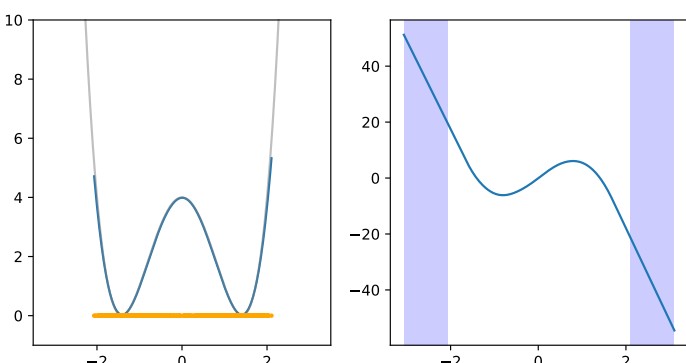

Figure 8: Illustration of the linear extension procedure. Left: the log target density (gray), the approximated value function at time step $t = t_{N-1}$ (blue), and the training samples used for approximation (orange). Right: the resulting approximate policy, with the shaded region indicating where the linear extension is activated.

In our experiments we use $q = 0.1$. We then introduce the adapted tensor domain at time step $t = t_n$ as

$$D_n := \bigtimes_{i=1}^{d} [a_{i,n}, b_{i,n}]. \tag{43}$$

Consequently, the tensor basis set from (12) is time step dependent, i.e. $\mathcal{B}_m = \mathcal{B}_m(D_n)$.

For the reverse sampling dynamics, trajectories might leave the considered evaluation domain $D_n$ at time $t = t_n$, rendering the evaluation of the gradient which enters the policy undefined, since the value function is approximated with basis functions from $\mathcal{B}_m(D_n)$. For this we consider a simple affine linear extension idea. Let $0 \le p < 1$ denote a domain shrinking factor and define the shrunken tensor domain

$$D_{n,p} = \bigtimes_{i=1}^{d} [a_{i,n,p}, b_{i,n,p}],$$

with

$$a_{i,n,p} = a_{i,n} + p|b_{i,n} - a_{i,n}|, \quad b_{i,n,p} = b_{i,n} - p|b_{i,n} - a_{i,n}|.$$

In our numerical experiment we used $p = 0.1$ in accordance to $q$ above.

Let $\boldsymbol{x} \in \mathbb{R}^d$. Then, we define the *extended gradient* evaluation of a function $v \in \operatorname{span} \mathcal{B}_m(D_n)$ as follows. Let $\Pi_{D_{n,p}} \boldsymbol{x}$ denote the projection of $\boldsymbol{x}$ onto $D_n$. Then

$$\nabla_{\mathrm{ext}} v(\boldsymbol{x}, t_n) := \nabla v(\Pi_{D_{n,p}} \boldsymbol{x}) + H_v(\boldsymbol{x} - \Pi_{D_{n,p}} \boldsymbol{x}),$$

with the well defined gradient $\nabla v$ and Hessian $H_v$ for points within $D_n$. In practice the computation of the Hessian will only be realized if $\boldsymbol{x} \ne \Pi_{D_{n,p}} \boldsymbol{x}$.

**Remark A.1.** We note that during the sampling stage, the vast majority of trajectories remain within the predefined moving domains $D_n$. As a result, only a small fraction of trajectories require evaluation of the Hessian, thereby limiting potential computational overhead. Moreover, as discussed above, the application of Hessians in the xTT format can in any case be carried out with high efficiency.

**Remark A.2.** The choice of $p > 0$ together with the design of an affine-type extension is motivated as follows. First, the affine structure ensures that samples leaving the computational domain are projected back while still being guided in an approximately correct direction. Second, potential boundary oscillations are absorbed in the exterior region parameterized by $q > 0$ and subsequently damped by $p > 0$.

An illustration of this extension mechanism is provided in Figure 8.

A.4.5   REPRESENTATION CHANGE ON DIFFERENT DOMAINS

In view of the iterative nature in Algorithm 1, multiple instances of minimization tasks of the explicit BSDE loss stated in (9) have to be solved. Any optimization task starts with an initial value. Motivated by a continuous change of the value function $V$ over time, a natural candidate for an initial guess at time step $t = t_n$ is the preliminary approximation of $V$ at time step $t = t_{n+1}$. However, since we enable a sequence of adaptively moving domains $(D_n)_n$, in general it holds $D_{n+1} = D_n$. Consequently, a plain copy paste of the extended tensor train information is not valid. In fact the basis functions at timestep $t = t_n$ will change, and in consequence the coefficient tensor.

Luckily, the extended tensor train format allows for an efficient change of representation between different domains, which we introduce as follows.

Let $D^1 = \times_{i=1}^d [a_{i,1}, b_{i,1}]$ and $D^2 = \times_{i=1}^d [a_{i,2}, b_{i,2}]$ be two tensor domains with non-trivial overlap $O = D^1 \cap D^2$. Then

$$O = \underset{i=1}{\overset{d}{\times}} [\max\{a_{i,1}, a_{i,2}\}, \min\{b_{i,1}, b_{i,2}\}] := \underset{i=1}{\overset{d}{\times}} [a_i, b_i]$$

is a tensor domain as well. Moreover for $\boldsymbol{m}^1 = (m_1^1, \ldots, m_d^1), \boldsymbol{m}^2 = (m_1^2, \ldots, m_d^2) \in \mathbb{N}^d$ consider two tensor basis sets with respect to $D^1$ and $D^2$:

$$\mathcal{B}_{\boldsymbol{m}^\ell}^\ell := \left\{ \phi_{\boldsymbol{\alpha}}^\ell := \bigotimes_{i=1}^d \phi_{\alpha_i}^{i,\ell} \,\middle|\, \boldsymbol{\alpha} \in [\boldsymbol{m}^\ell] \right\}, \quad \ell = 1, 2. \tag{44}$$

Note, that we omit the orthonormalization conditions at this point, as it is irrelevant for the discussion ahead.

We then follow a Galerkin projection type ansatz for the change of representation from basis functions defined in $D^1$ to basis function on $D^2$. Consider a tensor product Hilbert space $\mathcal{H}(O) = \bigotimes_{i=1}^d \mathcal{H}_i(a_i, b_i)$ with univariate inner product $(\cdot, \cdot)_{\mathcal{H}_i(a_i,b_i)}$ for $i = 1, \ldots, d$.

Let $i = 1, \ldots, d$ be fixed. Then we define the *overlap gramian matrix* for basis function from $\mathcal{B}_{\boldsymbol{m}^2}^2$ as

$$[G_i^2]_{jk} = (\phi_j^{i,2}, \phi_k^{i,2})_{\mathcal{H}_i(a_i,b_i)}, \qquad j, k = 1, \ldots, m_i^2, \tag{45}$$

and the *overlapping interaction matrix*

$$[M_i]_{jk} = (\phi_j^{i,2}, \phi_k^{i,1})_{\mathcal{H}_i(a_i,b_i)}, \qquad j, k = 1, \ldots, m_i^2. \tag{46}$$

Then, assuming that $G_i^2$ is invertible we define the univariate representation change operator

$$T_i := [G_i^2]^{-1} M_i \in \mathbb{R}^{m_i^2, m_i^1}. \tag{47}$$

*Application*: Now let $\widehat{V}_{\boldsymbol{C}^1}^1(\boldsymbol{x}) = \boldsymbol{C}^1[\Phi^1(\boldsymbol{x})]$ be an xTT $D^1$ with basis functions from $\mathcal{B}_{\boldsymbol{m}^1}^1$ and a tensor train given as

$$\boldsymbol{C}^1 = \boldsymbol{C}_1^1 \cdots \boldsymbol{C}_d^1.$$

Then, we define a new xTT

$$\widehat{V}_{\boldsymbol{C}^2}^2 = \boldsymbol{C}^2[\Phi^2(\boldsymbol{x})] \tag{48}$$

with

$$\boldsymbol{C}^2 = \boldsymbol{C}_1^2 \cdots \boldsymbol{C}_d^2, \quad \boldsymbol{C}_i^2 := T_i \circ_{2,2} \boldsymbol{C}_i^1, \tag{49}$$

that is $T_i$ acts on the 2nd slice of $\boldsymbol{C}_i^1$, responsible for the basis discretization.

In practice we choose $\mathcal{H}_i(O)$ according to the design of Hilbert spaces $\mathcal{H}_i(a_{i,1}, b_{i,1})$ and $\mathcal{H}_i(a_{i,2}, b_{i,2})$. For example if the latter is chosen to be $\mathcal{H}_i(a_{i,1}, b_{i,1}) = H^2(a_{i,1}, b_{i,1})$, then we choose the univariate Hilbert spaces on the directional overlaps $[a_i, b_i]$ to be Sobolev spaces of smoothness 2, i.e. $H^2(a_i, b_i)$. Then $\mathcal{H}(O) = H_{\mathrm{mix}}^2(O)$. Furthermore, the scalar products of (45) and (46) are computed via fast one-dimensional numerical integration at negligible cost.

### A.5 EMPIRICAL LOSS MINIMIZATION VIA ALS

We are now ready to formulate the alternating least square mechanism originated from Holtz et al. (2012a). As a first easier to access loss, we consider the classical empirical $L^2$ loss and then advance to our BSDE loss.

#### A.5.1 ALS FOR EMPIRICAL $L^2$ LOSSES

Let $D$ be a fixed tensor domain and assume we have samples $\boldsymbol{X}^{(k)} \in D$ and values $y^{(k)}$. Furthermore denote by $\boldsymbol{X}_i^{(k)}$, the $i$-th component of $\boldsymbol{X}^{(k)}$. Our goal is to fit a extended tensor train $v_{\boldsymbol{C}} \in \mathcal{X}_{\boldsymbol{R}}^{\boldsymbol{m}}$ to the data using a regularized empirical loss of the form

$$\mathcal{L}(v) := \frac{1}{K} \sum_{k=1}^{K} |v(\boldsymbol{X}^k) - y^{(k)}|^2 + \tau \|v\|_{\mathcal{H}}^2.$$

In the spirit of Appendix A.4.1 define the univariate feature matrices $W_i \in \mathbb{R}^{K, m_i}$ with

$$[W_i]_{k\ell} = \phi_\ell^i((\boldsymbol{X}^{(k)})_i), \qquad k = 1, \ldots, K, \ell = 1, \ldots, m_i. \tag{50}$$

Now assume the tensor train $\boldsymbol{C} = \boldsymbol{U}_1 \cdots \boldsymbol{U}_{i-1} \mathfrak{C} \boldsymbol{U}_{i+1} \cdots \boldsymbol{U}_d$ has its core at position $i$. Using subsequent contractions $\boldsymbol{U}_i \circ_{2,2} W_i \in \mathbb{R}^{r_{i-1}, K, r_i}$ and define $Y = (y^{(k)})_k \in \mathbb{R}^{K, 1}$ we can write

$$\mathcal{L}(v_{\boldsymbol{C}}) = \frac{1}{K} \sum_{k=1}^{K} |\boldsymbol{C}[\Phi(\boldsymbol{X}^k)] - y^{(k)}|^2 + \tau \|\boldsymbol{C}\|_{\mathrm{F}}^2. \tag{51}$$

$$= \frac{1}{K} \|\boldsymbol{U}_1 \circ_{2,2} W_1 \cdots \boldsymbol{U}_{i-1} \circ_{2,2} W_{i-1} \mathfrak{C} \circ_{2,2} W_i \cdots \boldsymbol{U}_d \circ_{2,2} W_d - Y\|_2^2 + \tau \|\mathfrak{C}\|_{\mathrm{F}}^2 \tag{52}$$

$$= \frac{1}{K} \|\boldsymbol{L}_i \mathfrak{C} \circ_{2,2} W_i \boldsymbol{R}_i - Y\|_2^2 + \tau \|\mathfrak{C}\|_{\mathrm{F}}^2, \tag{53}$$

with order three tensors (in fact matrices) given as

$$\boldsymbol{L}_i := \boldsymbol{U}_1 \circ_{2,2} W_1 \cdots \boldsymbol{U}_{i-1} \circ_{2,2} W_{i-1} \in \mathbb{R}^{1, K, r_{i-1}}, \tag{54}$$

$$\boldsymbol{R}_i := \boldsymbol{U}_{i+1} \circ_{2,2} W_{i+1} \cdots \boldsymbol{U}_d \circ_{2,2} W_d \in \mathbb{R}^{r_i, K, 1}. \tag{55}$$

Now the key observation is that $\boldsymbol{L}_i, \boldsymbol{R}_i$ and $W_i$ form a linear operator on $\mathfrak{C}$.

For this let $L_i \in \mathbb{R}^{K, r_{i-1}}$ and $R_i \in \mathbb{R}^{r_i, K}$ be the matrices representing the squeezed tensors $\boldsymbol{L}_i$ and $\boldsymbol{R}_i$, and $y \in \mathbb{R}^K$ the squeezed matrix $Y$, respectively. Then, we can define the matrix

$$A_i := R_i^\top \otimes W_i \otimes L_i \in \mathbb{R}^{K, r_{i-1} m r_i}. \tag{56}$$

Then, let $c := \mathrm{vec}(\mathfrak{C})$ and it holds

$$\mathcal{L}(v_{\boldsymbol{C}}) = \frac{1}{K} \|A_i c - y\|_2^2 + \tau \|c\|_2^2. \tag{57}$$

Then, in a sweeping manner forward and backward over each TT component, we shift the core to position $i$, then optimize the derived least square problem (57) and update the core with the reshape solution of the latter. The basic principle is summarized in Algorithm 2.

For the reader of interest, we mention that the subsequent shift of the core position in the for loop of Algorithm 2 (the so-called *forward-backward sweep*) only requires a single local singular value decomposition as described in Appendix A.2. In particular all other components in the current representation expect two (the former core and the shifted new core) stay the same. This observation allows for important speed gains of the algorithm, since the left component $\boldsymbol{L}_i$ and the right component $\boldsymbol{R}_i$ can be build during the backward sweep ($i = d, d-1, \ldots, 2$) and forward sweep ($i = 1, \ldots, d-1$) based on the principle of stacks as discussed in Appendix A.4. This in turn avoids same contractions during a single sweep.

As a error stop criteria, we choose $\epsilon > 0$ such that the change of the relative loss deviation

$$\frac{\sum_{k=1}^{K} |v(\boldsymbol{X}^k) - y^{(k)}|^2}{\|y\|_2^2}$$

---

**Algorithm 2** ALS for Tensor Train Minimization with Orthogonal Structure

---

1: **Input:** Initial components $\{C_1^{(0)}, \ldots, C_d^{(0)}\}$, regularization magnitude $\tau > 0$, max iterations $I$, error stop criteria $\epsilon$, data $(X^{(k)})_k, y \in \mathbb{R}^K$.
2: **Output:** Optimized TT representation $C = C_1 \cdots C_d$ for xTT $v_C \in \mathcal{X}_r^m$
3: Evaluate feature matrices $W_i$ from (50).
4: Set $n = 0$
5: **repeat**
6:     Set $n = n + 1$
7:     **for** $i = 1, 2, \ldots, d, d-1, \ldots 2$ **do**
8:         Set TT core to position $i$ yielding $C = U_1^{(n)} \cdots U_{i-1}^{(n)} \mathfrak{C} U_{i+1}^{(n)} \cdots U_d^{(n)}$
9:         Construct $A_i^{(n)}$ from left and right orthogonal components $L_i^{(n)}, R_i^{(n)}$ and $W_i$ as in (56).
10:        Solve micro-step problem:
11:             $c^* = \arg\min_{c \in \mathbb{R}^{r_{i-1}m_i r_i}} \|A_i^{(n)} c - y\|_2^2 + \tau \|c\|_F^2$
12:        Reshape $c^*$ to $\mathfrak{C}^* \in \mathbb{R}^{r_{i-1} \times m_i \times r_i}$ and update core at position $i$ to $\mathfrak{C} = \mathfrak{C}^*$.
13:     **end for**
14: **until** error stop criteria is met or $n \geq N$

---

is within $\epsilon$. This fraction is easily evaluated in each micro step at iteration $n$ and micro step $i$

$$\frac{1}{K} \sum_{k=1}^{K} |v(X^k) - y^{(k)}|^2 = \frac{1}{K} \|A_i^{(n)} \operatorname{vec}(\mathfrak{C}) - y\|_2^2.$$

### A.5.2 ALS FOR BSDE LOSSES WITH GENERAL BASIS FUNCTIONS

We now extend the ALS approach to the BSDE loss function for the general case where the derivatives of basis functions are not necessarily in the span of the original basis functions. This occurs, for example, with $\mathcal{C}^s$ splines of polynomial degree $p \geq s + 2$.

Our starting point is the BSDE loss:

$$\widehat{\mathcal{L}}_n^K(\widehat{V}_n) = \frac{1}{K} \sum_{k=1}^{K} \left( (\mathrm{I} + \langle \widehat{\Sigma}^{n,(k)}, \nabla \rangle) \widehat{V}^n(\widehat{X}_n^{u,(k)}) - y_{n+1}^{(k)} \right)^2. \tag{58}$$

Let $\widehat{V}^n$ be represented by an xTT $v_C \in \mathcal{X}_r^m$ with tensor train with core position at $j$ given as

$$C = U_1 \cdots U_{j-1} \mathfrak{C} U_{j+1} \cdots U_d.$$

For each sample $k = 1, \ldots, K$, define the basis evaluations:

$$w_j^{(k)}[\alpha_j] := \phi_{\alpha_j}^j(X_{n,j}^{u,(k)}),$$
$$\dot{w}_j^{(k)}[\alpha_j] := \partial_x \phi_{\alpha_j}^j(X_{n,j}^{u,(k)}), \quad \forall j = 1, \ldots, d,$$

where we omit the dependency on $u$ and $n$ for easier readability. We introduce the feature matrices $W_i, \dot{W}_i \in \mathbb{R}^{K, m_i}$ with

$$[W_i]_{k\ell} = w_i^{(k)}[\ell], \qquad [\dot{W}_i]_{k\ell} = \partial_{x_i} \dot{w}_i^{(k)}[\ell] \tag{59}$$

Introduce the weighted feature matrices for $i = 0, \ldots, d$ and $j = 1, \ldots, d$:

$$W_{ij} = \begin{cases} W_j & \text{if } i \neq j, \\ \widehat{\Sigma}_j^n * \dot{W}_j & \text{if } i = j, \end{cases} \tag{60}$$

where the operator $*$ refers to pointwise multiplication in the sample direction slice for each $k = 1, \ldots, K$ with the vector $\widehat{\Sigma}_i^n \in \mathbb{R}^K$ with

$$(\widehat{\Sigma}_i^n)_k = [\widehat{\Sigma}^{n,(k)}]_i.$$

This allows us to define rank 1 tensors:

$$\boldsymbol{W}_0 = W_{0,1} \otimes \cdots \otimes W_{0,d},$$
$$\dot{\boldsymbol{W}}_i = W_{i,1} \otimes \cdots \otimes W_{i,d}.$$

Now, with the core at position $j$, we define the left and right contractions for each $i = 0, \ldots, d$:

$$\boldsymbol{L}_{i,j} = \boldsymbol{U}_1 \circ_{2,2} W_{i1} \cdots \boldsymbol{U}_{j-1} \circ_{2,2} W_{i,j-1} \in \mathbb{R}^{1,K,r_{i-1}}$$
$$\boldsymbol{R}_{i,j} = \boldsymbol{U}_{j+1} \circ_{2,2} W_{i,j+1} \cdots \boldsymbol{U}_d \circ_{2,2} W_{i,d} \in \mathbb{R}^{r_i,K,1}$$

with the conventions $L_{i,1} = \boldsymbol{1}$ and $R_{i,d} = \boldsymbol{1}$.

The BSDE operator applied to the tensor train becomes for $(X_{n,j}^{u_\ell,(k))_k} \in \mathbb{R}^{K,d}$:

$$(\mathrm{I} + \langle \widehat{\Sigma}^{n,(k)}, \nabla \rangle) \widehat{V}^n ((X_{n,j}^{u_\ell,(k))_k})$$

$$= \boldsymbol{L}_{0,j}(\mathfrak{C} \circ_{2,2} W_{0j}) \boldsymbol{R}_{0,j} + \sum_{i=1}^{d} \boldsymbol{L}_{i,j}(\mathfrak{C} \circ_{2,2} W_{ij}) \boldsymbol{R}_{i,j}$$

$$= \sum_{i=0}^{d} \boldsymbol{L}_{i,j}(\mathfrak{C} \circ_{2,2} W_{ij}) \boldsymbol{R}_{i,j}.$$

Analogously to the derivation in (53) for vector $y = (y_{n+1}^{(k)})_k \in \mathbb{R}^{K,1}$ we obtain

$$\widehat{\mathcal{L}}_n^K(v_{\boldsymbol{C}}) = \frac{1}{K} \left\| \sum_{i=0}^{d} \boldsymbol{L}_{i,j}(\mathfrak{C} \circ_{2,2} W_{ij}) \boldsymbol{R}_{i,j} - y \right\|_2^2 + \tau \|\mathfrak{C}\|_{\mathrm{F}}^2, \tag{61}$$

Recall at this point that $\boldsymbol{L}_{i,j}, W_{i,j}$ and $\boldsymbol{R}_{i,j}$ depend on $u$ and $n$. Again we can define the matrix

$$A_j := \sum_{i=0}^{d} R_{i,j}^\top \otimes W_{i,j} \otimes L_{i,j} \in \mathbb{R}^{K,r_{i-1}mr_i}, \tag{62}$$

where $R_{i,j}$ and $L_{i,j}$ again denote the matrices obtained from squeezing the last mode dimension of $\boldsymbol{R}_{i,j}$ and the first mode dimension of $\boldsymbol{L}_{i,j}$.

Then in the spirit of (56) it holds that

$$\widehat{\mathcal{L}}_n^K(v_{\boldsymbol{C}}) = \frac{1}{K} \|A_i \operatorname{vec}(\mathfrak{C})\|_2^2 + \tau \| \operatorname{vec}(\mathfrak{C})\|_2^2 \tag{63}$$

The computational efficiency is maintained through the stack principle: during the forward sweep ($j = 1, \ldots, d$), the left contractions $L_{i,j}^{(k)}$ can be updated incrementally, and during the backward sweep ($j = d, \ldots, 2$), the right contractions $R_{i,j}^{(k)}$ can be updated incrementally.

## A.6 ADAPTIVITY

Our algorithm features adaptivity with respect to the regularization parameter $\tau_n$ (see Appendix A.6.1), the used basis degrees and the underlying xTT rank (see Appendix A.6.2) used for the functional tensor train approximation.

### A.6.1 CHOICE OF REGULARIZATION MAGNITUDE $\tau_n$

Recall that our regularized loss in (16) is given as

$$\widehat{\mathcal{L}}_n^K(\widehat{V}_{\boldsymbol{C}}) + \tau_n \|\widehat{V}_{\boldsymbol{C}}\|_{\mathcal{H}}^2 = \frac{1}{K} \sum_{k=1}^{K} \left( \boldsymbol{C}[\Phi(\widehat{X}_n^u)] + \Sigma_n^{(k)} \cdot \nabla \boldsymbol{C}[\Phi(\widehat{X}_n^u)] - y_{n+1}^{(k)} \right)^2 + \tau_n \|\boldsymbol{C}\|_{\mathrm{F}}^2. \tag{64}$$

If the tensor train $\boldsymbol{C}$ is represented as

$$\boldsymbol{C} = \boldsymbol{C}_1 \boldsymbol{C}_2 \cdots \boldsymbol{C}_d = \boldsymbol{U}_1 \cdots \boldsymbol{U}_{i-1} \mathfrak{C} \boldsymbol{U}_{i+1} \cdots \boldsymbol{U}_d$$

---

**Algorithm 3** ALS for BSDE with General Basis Functions

---

1: **Input:** Initial TT cores $\{U_1^{(0)}, \ldots, U_d^{(0)}\}$, regularization $\tau > 0$, max iterations $I$, tolerance $\epsilon$, samples $(\widehat{X}_n^{u,(k)}, \Sigma_n^{(k)}, y_{n+1}^{(k)})_k$

2: **Output:** Optimized TT representation for $\widehat{V}_n$

3: Precompute feature evaluation matrices $W_j, \dot{W}_j$ from (59) and define $W_{ij}$ from (60).

4: Set $n = 0$

5: **repeat**

6:    Set $n = n + 1$

7:    **for** $j = 1, 2, \ldots, d, d-1, \ldots, 2$ **do**

8:       Set TT core to position $j$: $\boldsymbol{C} = \boldsymbol{U}_1^{(n)} \cdots \boldsymbol{U}_{j-1}^{(n)} U_j \boldsymbol{U}_{j+1}^{(n)} \cdots \boldsymbol{U}_d^{(n)}$

9:       Compute $\boldsymbol{L}_{i,j}, \boldsymbol{R}_{i,j}$ using pre stack principle for each $i = 0, \ldots, d$

10:      Build micro step system matrix $A_j$ from (62)

11:      Solve: $c^* = \arg\min_c \|A_j c - y\|_2^2 + \tau \|c\|_F^2$

12:      Reshape $c^*$ to update core $\mathfrak{C}$

13:    **end for**

14:    Compute loss: $\mathcal{L}^{(n)} = \frac{1}{K}\|A_j c - b\|_2^2$

15: **until** $|\mathcal{L}^{(n)} - \mathcal{L}^{(n-1)}| < \epsilon$ or $n \geq I$

---

with orthogonal order three tensors $U_j \in \mathbb{R}^{r_{j-1}, m_j, r_j}$ for $j = 1, \ldots, i-1, i+1, \ldots d$ and a *core* $\mathfrak{C} \in \mathbb{R}^{r_{i-1}, m_i, r_i}$ at position $i$, then, we have that

$$\widehat{\mathcal{L}}_n^K(\widehat{V}_{\boldsymbol{C}}) + \tau_n \|\widehat{V}_{\boldsymbol{C}}\|_{\mathcal{H}}^2 = \|A_{i,n}^u \operatorname{vec}(\mathfrak{C}) - y_{n+1}\|_2^2 + \tau_n \|\mathfrak{C}\|_F^2, \tag{65}$$

for a matrix $A_{i,n}^u \in \mathbb{R}^{K, r_{i-1} m_i r_i}$. In particular, the original empirical BSDE loss is easy to evaluate in each micro step of our proposed ALS algorithm with

$$\widehat{\mathcal{L}}_n^K(\widehat{V}_{\boldsymbol{C}}) = \|A_{i,n}^u \operatorname{vec}(\mathfrak{C}) - y_{n+1}\|_2^2$$

and the regularization in the $\mathcal{H}$ norm reduces through Parseval identity and the core representation of the tensor train to a single Frobenius norm evaluation of the core $\mathfrak{C}$. In our experiments we found that a naive choice of $\tau_n$, e.g. $\tau_n \equiv 10^{-8}$ for all $n$ independent of the dimension $d$, did not yield a convergent scheme. This can be understood via the Hilbert space norm $\|\widehat{V}_{\boldsymbol{C}}\|_{\mathcal{H}}^2$. Note that our goal is that at time step $t = t_n$ the minimizer of the above loss, say $\widehat{V}_{\boldsymbol{C}}$, should approximate the true value function $V(\cdot, t_n)$. However, the function

$$t \mapsto \|V(\cdot, t)\|_{\mathcal{H}}^2$$

can vary over time and over dimension $d$. In fact, in our experiments the function seems to scale exponentially in $d$. Consequently, $\tau_n$ has to absorb the effect of the strength of the $\|\cdot\|_{\mathcal{H}}^2$ term. In our experiments, we set the regularization strength to $\gamma = 0.1$ and determine $\tau_n$ as follows. At time step $n = N$ we set an initial guess of $\tau_N$. Assume at time $t = t_n$ in the $k$-th sweep at micro step $i$, we obtained an update of the tensor train $\boldsymbol{C}$, in particular for the core at position $i$ using the current value $\tau_n$. This allows us to compute $\|A_{i,n}^u \operatorname{vec}(\mathfrak{C}) - y_{n+1}\|_2^2$ and $\|\mathfrak{C}\|_F^2$. We then update

$$\tau_n = \gamma \frac{\|A_{i,n}^u \operatorname{vec}(\mathfrak{C}) - y_{n+1}\|_2^2}{\|\mathfrak{C}\|_F^2}$$

and go to the next micro step. This ansatz ensures that

$$\tau_n \|\mathfrak{C}\|_F^2 = \gamma \|A_{i,n}^u \operatorname{vec}(\mathfrak{C}) - y_{n+1}\|_2^2,$$

i.e. it is a magnitude of order $\gamma$ smaller than the loss of interest. In our experiments we use $\gamma = 0.1$. Then, when going from timestep $t = t_{n+1}$ to time step $t = t_n$, the preliminarily found $\tau_{n+1}$ is a natural candidate for the initial value of $\tau_n$, since $\|V(\cdot, t_n)\|_{\mathcal{H}}^2 \approx \|V(\cdot, t_{n+1})\|_{\mathcal{H}}^2$.

It is worth noting that choosing $\gamma = 0$, i.e. $\tau_n = 0$, does not yield a robust algorithm, since oscillations in the derivatives still have to be controlled. In our experiments we found that very small magnitudes of $\tau_n$ were necessary to stabilize our learning algorithm, see Figure 9.

**Remark A.3.** An alternative approach would be to construct the tensor basis orthonormally with respect to the (empirical) weighted Hilbert spaces, which corresponds to orthonormalization using empirical Grammians. This is expected to yield better control over the magnitude of $|\cdot|_{\mathcal{H}}^2$, thereby reducing the algorithm's sensitivity to the choice of $\tau_n$. We leave this direction for future work.

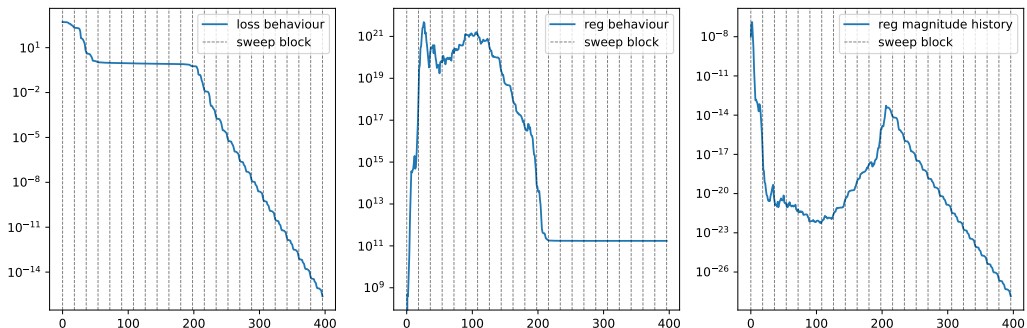

Figure 9: Example of adaptively finding the refularization parameter $\tau_n$ for the Multiwell problem in dimension $d = 10$. Left: The empirical loss value $\widehat{\mathcal{L}}_n^K(\widehat{V}_C)$. Middle: The value $\|\widehat{V}_C\|_{\mathcal{H}}^2$. Right: The adaptive update of $\tau_N$.

### A.6.2 ADAPTIVITY FOR RANK DESIGN

An important feature of our algorithm involves the adaptive rank selection, e.g. choosing proper TT ranks required for a rigorous approximation quality of the value function. Based on the design of the HJB PDE (4) we expect that the value function $V(\cdot, t)$ has varying FTT rank over time $t$. Intuitively, we expect the nonlinear term $\|\sigma^\top \nabla V\|^2$ to lead to a short time range of rank increase while the diffusion term over time will push the FTT rank $\overline{r} \in \mathbb{N}^{d-1}$ to the TT rank of the log-density of the invariant distribution of the reversing process. For example, if the reversing process is a simple OU process, then the invariant distribution is standard normal and the limit FTT rank will be $\mathbf{2} = (2, \ldots, 2) \in \mathbb{N}^{d-1}$, see Gruhlke et al. (2025).

We recall the SVD of a tensor train component $\boldsymbol{C}_i$ after reshaping to a matrix $C_i$ from Appendix A.2, that reads

$$C_i = U_i \Sigma_i V_i^\top.$$

The decay pattern of the singular values in $\Sigma_i$, denoted $\sigma_1^i, \ldots, \sigma_{r_i}^i$, provides valuable information for adapting the rank $r_i$. Specifically, if all singular values are of comparable and non-negligible magnitude, a higher rank $r_i$ may be warranted. Conversely, if the tail singular values $\sigma_{k+1}^i, \ldots, \sigma_{r_i}^i$ are close to zero, the effective rank can be safely reduced to $r_i = k$. It is also sensible to assess the relative magnitude of the singular values against the scale of the loss function, as this helps mitigate overfitting to sampling noise.

Now, we can apply high-order singular value decomposition (HOSVD, see Grasedyck (2010); Oseledets (2011)) on $\boldsymbol{C}$. To this end, let us fix $1 \le k \le d$ as the so-called *core position*. Then for any $i = 1, \ldots, k-1$, we iteratively reshape the $i$-th TT component $\boldsymbol{C}_i \in \mathbb{R}^{r_{i-1}, m_i, r_i}$ as a matrix $C_i \in \mathbb{R}^{r_{i-1}, m_i r_i}$, apply a singular value decomposition

$$C_i = U_i \Sigma_i V_i^\top,$$

and contract the non-orthonormal part $\Sigma_i V_i^\top$ from left to $\boldsymbol{C}_{i+1}$ and redefine

$$\boldsymbol{C}_{i+1} = \Sigma_i V_i^\top \boldsymbol{C}_{i+1}.$$

Analogously, for $i = d, d-1, \ldots, k+1$ we iteratively perform the SVDs $C_i = U_i \Sigma_i V_i^\top$ and contract $U_i \Sigma_i$ from right to $C_{i-1}$. Finally, we can reshape each $U_i \in \mathbb{R}^{r_{i-1}, m_i, r_i}$ to an order three tensor $\boldsymbol{U}_i \in \mathbb{R}^{r_{i-1}, m_i, r_i}$ for $i = 1, \ldots, k-1$ and $V_i^\top \in \mathbb{R}^{r_{i-1}, m_i, r_i}$ to an order three tensor $\boldsymbol{U}_i \in \mathbb{R}^{r_{i-1}, m_i, r_i}$ for $i = d, d-1, \ldots, k+1$. Then, it holds that

$$\boldsymbol{C} = \boldsymbol{U_1} \cdots \boldsymbol{U}_{k-1} \boldsymbol{C}_k \boldsymbol{U}_{k+1} \cdots \boldsymbol{U}_d. \tag{66}$$

We propose a simple adaptive rank design in Algorithm 4.

Additionally, rank adjustments can be guided by target loss values, taking into account the level of sample noise. In our experiments, we did not employ this approach. Since the algorithm requires multiple optimization instances per time step $t$, we introduce a rank update frequency to regulate the application of Algorithm 4.

---

**Algorithm 4** Adaptive Rank Adjustment for Tensor Train Learning

---

1: **Input:** Initial TT-rank $\boldsymbol{r}^{(0)} = (r_1^{(0)}, \ldots, r_{d-1}^{(0)})$, threshold $\delta > 0$, maximum iterations $N$
2: **Output:** Optimized TT representation with adapted ranks
3: Initialize $\rho_i \leftarrow \emptyset$ for $i = 1, \ldots, d-1$               {Set of previously seen ranks}
4: fixed $\leftarrow \emptyset$                                             {Set of fixed rank indices}
5: **repeat**
6:     Optimize loss with current fixed TT-rank $\boldsymbol{r}$              {Using ALS or other method}
7:     **for** $i = 1$ to $d-1$ **do**
8:        **if** $i \notin$ fixed **then**
9:           Perform HOSVD and get singular values $\sigma_1^i \geq \cdots \geq \sigma_{r_i}^i$
10:           $k_i \leftarrow \min\{\ell \mid \sigma_\ell^i < \delta\sigma_1^i\}$
11:           **if** $k_i = r_i$ **then**
12:              $r_i^{\text{new}} \leftarrow r_i + 1$                                 {Increase rank}
13:              Add random orthonormal direction to $U_i$ and $U_{i+1}$
14:           **else if** $k_i < r_i$ **then**
15:              $r_i^{\text{new}} \leftarrow k_i$                                     {Decrease rank}
16:           **end if**
17:           **if** $r_i^{\text{new}} \in \rho_i$ **then**
18:              fixed $\leftarrow$ fixed $\cup \{i\}$                             {Fix this rank}
19:           **else**
20:              $\rho_i \leftarrow \rho_i \cup \{r_i^{\text{new}}\}$
21:              $r_i \leftarrow r_i^{\text{new}}$
22:           **end if**
23:        **end if**
24:     **end for**
25: **until** fixed $= \{1, \ldots, d-1\}$ or maximum iterations reached

---

In our experiments, we typically set the rank update frequency to 10 or 20 and choose the relative singular value magnitude parameter as $\delta = 10^{-4}$.

### A.6.3 ADAPTIVITY FOR BASIS DEGREES

The third adaptivity component of our algorithm involves adaptivity with respect to the choice of basis degrees needed. Based on the design of the HJB PDE (4), the initial condition $V(\cdot, 0)$ and the final negative log-density $V(\cdot, T) \approx -\log \mathcal{N}(0, I)$ in general require different sizes of spatial discretization spaces. Moreover, for each $t \in (0, T)$ the function $V(\cdot, t)$ is expected to be a smooth function that only requires adapted spatial discretizations for different time steps $t$.

Hence, adapted spatial discretization yields two different advantages:

- Increased learning speed: Reducing the number of degrees of freedom in each xTT reduces the workload in the minimization step, e.g. solving (16).

- The reduced basis degree has a stabilization effect: Reduced sample complexity improves the conditioning of the local least square problems, and the numerical noise in the coefficients of unnecessary basis degrees will be propagated and will thus not introduce additional error propagation.

The idea is based on a simple heuristic. Fix a given sparsity gap integer $s \in \mathbb{N}$ (typically $s = 1, 2$) and a loss tolerance $\text{tol}_{\text{loss}}$. Then, at iteration step $k$ optimize the loss for a given xTT with rank $\boldsymbol{r}^{(0)}$ and basis degree $\boldsymbol{m}^{(0)} = (m_1^{(0)}, \ldots, m_d^{(0)})$.

- *Basis degree reduction:* Assume the optimized loss $\mathcal{L}_{\min}^{(0)}$ obtained by solving (16) is below the tolerance $\text{tol}_{\text{loss}}$. Then, reduce $\boldsymbol{m}^{(0)}$ as much as possible to obtain $\boldsymbol{m}^{(1)} \leq \boldsymbol{m}^{(0)}$, such that the associated minimized loss $\mathcal{L}_{\min}^{(1)}$ using basis degrees $\boldsymbol{m}^{(1)}$ remains below the tolerance threshold $\text{tol}_{\text{loss}}$. Choose $\boldsymbol{m}^{(1)}$ as the new basis degree and proceed to the next iteration $k+1$.

- *Basis degree increasing:* Choose a limit value $L \in \mathbb{N}$. Also, assume the optimized loss $\mathcal{L}_{\min}^{(0)}$ obtained by solving (16) is above the tolerance $\text{tol}_{\text{loss}}$. Then, increase $\boldsymbol{m}^{(0)}$ component-wise,

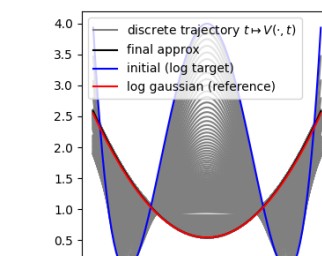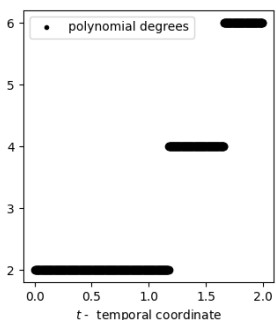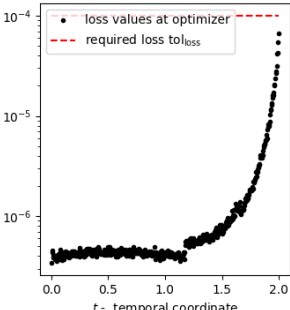

Figure 10: Illustration of basis-degree adaptivity for a one-dimensional example. Left: Trajectory transformation from the log-target distribution (log multimodal) to the log-prior distribution (log Gaussian). Middle: The selected polynomial degree decreases when moving from right ($t = T = 2$) to left ($t = 0$). In the limit $t \to 0$, the solution approaches a log-Gaussian, which requires only a degree of 2. Legendre polynomials are used as basis functions in this experiment. Right: Loss error and corresponding loss tolerance.

leading to $\boldsymbol{m}^{\ell+1} \geq \boldsymbol{m}^\ell$, for $\ell = 0, 1, \ldots, L$ until the associated minimized loss $\mathcal{L}_{\min}^{(\ell)}$ using basis degrees $\boldsymbol{m}^{(\ell)}$ is below the tolerance threshold or if the latest $s$ steps do not yield a suitable improvement in the loss reduction and consider rank adaptivity, see Remark A.4.

**Remark A.4.** The sparsity gap becomes relevant when the chosen basis contains components that do not contribute meaningfully to the representation of the target function. For example, consider the function $f(x) = L_2(x) + L_4(x)$, where $L_i$ denotes the Legendre polynomials orthonormal in $L^2(-1, 1)$. Suppose we approximate $f$ using the basis $\{L_0, L_1, L_2\}$. In this case, enlarging the basis by adding $L_3$ – and thereby increasing the polynomial degree – does not reduce the projection error, since $L_3$ has no overlap with the nonzero components of $f$.

More generally, when increasing the basis degree no longer yields a substantial reduction in the loss, it becomes preferable to increase the xTT rank, as described in Algorithm 4.

To illustrate the principle underlying our degree-adaptivity strategy, we consider a one-dimensional example, i.e., the multi-modal setup from Section 4 with $d = 1$. In this setting, there is no rank interaction, allowing us to isolate the effect of basis-degree adaptivity. Figure 10 demonstrates this behavior in practice: for $T = 2$, the algorithm selects a polynomial degree of 6 in the initial optimization step of the BSDE loss, after which the degree gradually decreases as $t \to 0$.

### A.7 EXPLICIT FUNCTIONAL TENSOR TRAIN RANKS

This section is devoted to the analysis of explicit FTT ranks for solutions of the HJB equation. It is well known that the Ornstein–Uhlenbeck process with a standard Gaussian invariant distribution, when initialized with a general Gaussian distribution, remains Gaussian throughout its evolution. Consequently, the associated HJB equation (see Lemma 2.1) admits, for every $t$, a solution $V(\cdot, t)$ that is a log-Gaussian potential.

For this reason, let us consider general Gaussian potentials of the form

$$x \mapsto x^\top M x,$$

where $M$ is a symmetric and positive definite matrix.

It turns out that we can explicitly state the FTT ranks of these potentials, summarized in Theorem A.5 and Corollary A.6. This discussion is inspired by Gruhlke et al. (2025).

At the level of the potential, low-rank subdiagonal blocks have a very clear implication for the FTT ranks. We define the $i$-th subdiagonal block of $M$, for $i \in \{1, \ldots, d-1\}$, as the matrix $M_{1:i,i+1:d} \in \mathbb{R}^{i \times (d-i)}$ in the following block decomposition:

$$M = \begin{bmatrix} M_{1:i,1:i} & M_{i+1:d,1:i} \\ M_{1:i,i+1:d} & M_{i+1:d,i+1:d} \end{bmatrix}.$$

As the following theorem shows, the ranks of the subdiagonal blocks of $M$ fully determine the FTT rank of $x^\top M x$.

**Theorem A.5** (FTT rank bounds for Gaussians with low-rank subdiagonal blocks). *Let $d \in \mathbb{N}$ and $f : \mathbb{R}^d \to \mathbb{R}$ admit the form $f(x) = x^\top M x$ for a symmetric invertible matrix $M \in \mathbb{R}^{d,d}$. Furthermore, assume $M$ has sub-diagonal blocks $M_{1:i,i+1:d}$, $i = 1, \ldots, d-1$ with ranks given by $\ell_i \in \mathbb{N}$. Then $f$ has finite FTT rank $\mathbf{r} \in \mathbb{N}^{d-1}$. In particular for $d \geq 3$,*

$$\mathbf{r} = 2 + (\ell_1, \ell_2, \ldots, \ell_{d-1})$$

*and $\mathbf{r} = 2 \in \mathbb{N}$ for $d = 2$. In particular, in the case of an isotropic Gaussian, we have $\mathbf{r} \equiv 2$.*

*Proof.* First, note that there is only something to prove if the rank bounds $\ell_i$ satisfy

$$\ell_i \leq i + 2, \qquad \text{if } i \leq \lfloor \tfrac{d}{2} \rfloor,$$
$$\ell_i \leq d - i + 2, \qquad \text{if } i > \lfloor \tfrac{d}{2} \rfloor,$$

as the TT rank bounds will be higher than the maximal ranks otherwise. So let $i \in \{1, \ldots, d-1\}$ and $\ell_i$ satisfying the respective condition be the rank of the subdiagonal block $M_{i+1:d,1:i}$, which is defined by

$$M_{i+1:d,1:i} = \begin{pmatrix} m_{i+1,1} & \cdots & m_{i+1,i} \\ \vdots & & \vdots \\ m_{d,1} & \cdots & m_{d,i} \end{pmatrix}.$$

We consider a singular value decomposition of $M_{i+1:d,1:i}$ into

$$M_{i+1:d,1:i} = \underbrace{U}_{\in \mathbb{R}^{* \times \ell_i}} \underbrace{\Sigma}_{\in \mathbb{R}^{\ell_i \times \ell_i}} \underbrace{V^T}_{\in \mathbb{R}^{\ell_i \times *}}.$$

By construction, all terms in $x^T M x$ including mixtures of the variables indexed in $\{1, \ldots, i\}$ with those indexed in $\{i+1, \ldots, d\}$ are given by

$$2 \begin{pmatrix} x_{i+1} & \cdots & x_d \end{pmatrix} U \Sigma V^T \begin{pmatrix} x_1 \\ \vdots \\ x_i \end{pmatrix} = 2 \left[ U^T \begin{pmatrix} x_{i+1} \\ \vdots \\ x_d \end{pmatrix} \right]^T \Sigma V^T \begin{pmatrix} x_1 \\ \vdots \\ x_i \end{pmatrix}.$$

Hence, we have

$$\begin{aligned}
f(x) &= \begin{pmatrix} 1 & 2x_{1:i}^T & x_{1:i}^T \cdot M_{1:i,1:i} \cdot x_{1:i} \end{pmatrix} \begin{pmatrix} x_{i+1:d}^T \cdot M_{i+1:d,i+1:d} \cdot x_{i+1:d} \\ M_{1:i,i+1:d} \cdot x_{i+1:d} \\ 1 \end{pmatrix} \\
&= \begin{pmatrix} 1 & 2x_{1:i}^T & x_{1:i}^T \cdot M_{1:i,1:i} \cdot x_{1:i} \end{pmatrix} \begin{pmatrix} 1 & 0 & 0 \\ 0 & M_{1:i,i+1:d} & 0 \\ 0 & 0 & 1 \end{pmatrix} \begin{pmatrix} x_{i+1:d}^T \cdot M_{i+1:d,i+1:d} \cdot x_{i+1:d} \\ x_{i+1:d} \\ 1 \end{pmatrix} \\
&= \begin{pmatrix} 1 & 2x_{1:i}^T & x_{1:i}^T \cdot M_{1:i,1:i} \cdot x_{1:i} \end{pmatrix} \begin{pmatrix} 1 & 0 & 0 \\ 0 & V\Sigma U^T & 0 \\ 0 & 0 & 1 \end{pmatrix} \begin{pmatrix} x_{i+1:d}^T \cdot M_{i+1:d,i+1:d} \cdot x_{i+1:d} \\ x_{i+1:d} \\ 1 \end{pmatrix} \qquad (67) \\
&= \begin{pmatrix} 1 & 2x_{1:i}^T \cdot V & x_{1:i}^T \cdot M_{1:i,1:i} \cdot x_{1:i} \end{pmatrix} \begin{pmatrix} 1 & 0 & 0 \\ 0 & \Sigma & 0 \\ 0 & 0 & 1 \end{pmatrix} \begin{pmatrix} x_{i+1:d}^T \cdot M_{i+1:d,i+1:d} \cdot x_{i+1:d} \\ U^T \cdot x_{i+1:d} \\ 1 \end{pmatrix} \\
&= \underbrace{\begin{pmatrix} 1 & 2x_{1:i}^T \cdot V & x_{1:i}^T \cdot M_{1:i,1:i} \cdot x_{1:i} \end{pmatrix}}_{\in \mathbb{R}^{1 \times (\ell_i + 2)}} \underbrace{\begin{pmatrix} x_{i+1:d}^T \cdot M_{i+1:d,i+1:d} \cdot x_{i+1:d} \\ \Sigma U^T \cdot x_{i+1:d} \\ 1 \end{pmatrix}}_{\in \mathbb{R}^{(\ell_i + 2) \times 1}}.
\end{aligned}$$

Since we have completely separated the first $i$ variables from the last $d - i$, this proves that the $i$-th rank is bounded by $\ell_i + 2$. Conversely, the rank can not be lower than $\ell_i + 2$ as that would be in violation to $M_{i+1:d,1:i}$ having rank $\ell_i$. $\qquad \square$

The above result yields a worst case estimate of the ranks in the case that no structure of the precision matrix $M$ is known.

**Corollary A.6** (FTT rank bound for Gaussian potentials: worst case). *Let $d \in \mathbb{N}$ and $\Phi \colon \mathbb{R}^d \to \mathbb{R}$ admit the form $\Phi(x) = x^\top M x$ for a symmetric invertible matrix $M \in \mathbb{R}^{d,d}$. Then $\Phi$ has finite FTT rank $\boldsymbol{r} \in \mathbb{N}^{d-1}$. In particular for $d \geq 3$,*

$$\boldsymbol{r} \leq \overline{\boldsymbol{r}} := 2 + \begin{cases} \left(1, 2, \ldots, \frac{d}{2}, \ldots, 2, 1\right), & d \text{ even,} \\ \left(1, 2, \ldots, \frac{d-1}{2}, \frac{d-1}{2}, \ldots, 2, 1\right), & d \text{ odd,} \end{cases}$$

*and $\boldsymbol{r} = 2 \in \mathbb{N}$ for $d = 2$.*

*Proof.* Direct consequence of the sub-diagonal rank bounds $\ell_i$ of $M$ by application of Theorem A.5. $\square$

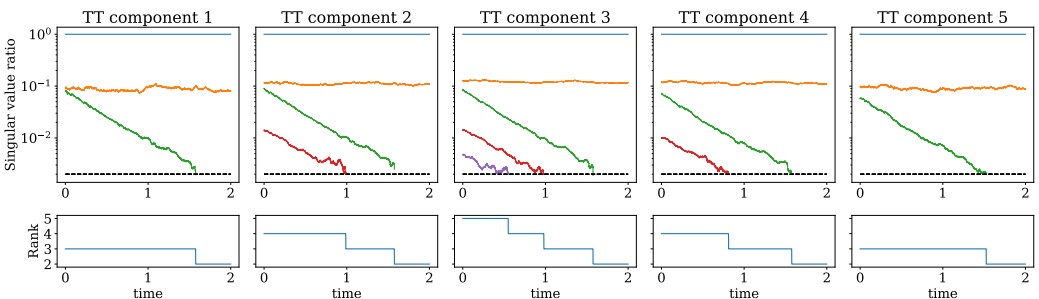

Figure 11: We display the singular values of all tensor train components for an example in dimension $d = 6$ over time and reduce the rank whenever a corresponding value is below a prespecified threshold.

# B  NUMERICAL DETAILS

## B.1  LIMITATIONS

While the outer iterations of our algorithm encourage the discovery of new modes (see Figure 1), we note that there is no guarantee that all modes of the target distribution will be identified – although this limitation is shared by all existing sampling algorithms. A further restriction arises from the requirement that the value function $V$ admits a sufficiently low-rank representation, which may not hold for all target distributions. Finally, although we have substantially improved the stability of a naive implementation of Algorithm 1 combined with FTTs (see Remark 3.3 and our experimental results), certain target distributions may still pose additional challenges, particularly due to the expressive and numerical limitations of FTTs.

## B.2  EVALUATION METRICS

In this section, we define the evaluation metrics used to measure sampling quality. Our metrics are based on path weights

$$w = \frac{\mathrm{d}\mathbb{P}_{\breve{Y}}}{\mathrm{d}\mathbb{P}_{X^u}}(X^u) \approx \frac{p_{\text{target}}(\widehat{X}_N^u) \prod_{n=0}^{N-1} \breve{p}_{n|n+1}(\widehat{X}_n^u | \widehat{X}_{n+1}^u)}{p_{\text{prior}}(\widehat{X}_0^u) \prod_{n=0}^{N-1} \vec{p}_{n+1|n}(\widehat{X}_{n+1}^u | \widehat{X}_n^u)}, \tag{68}$$

where $\widehat{X}^u$ is the discretized SDE, as defined in (8) (Richter & Berner, 2024; Blessing et al., 2024). Here, $\vec{p}_{n+1|n}$ and $\breve{p}_{n|n+1}$ are the forward and backward transition densities of the discrete forward and backward process, respectively, given by

$$\breve{p}_{n|n+1}(\widehat{X}_n^u | \widehat{X}_{n+1}^u) = \mathcal{N}\left(\widehat{X}_n | \widehat{X}_{n+1} - f(\widehat{X}_{n+1}, (n+1)\Delta t)\Delta t, \sigma^2((n+1)\Delta t)\Delta t\right) \tag{69}$$

$$\vec{p}_{n+1|n}(\widehat{X}_{n+1}^u | \widehat{X}_n^u) = \mathcal{N}\left(\widehat{X}_{n+1} | \widehat{X}_n + (f + \sigma u)(\widehat{X}_n, n\Delta t)\Delta t, \sigma^2(n\Delta t)\Delta t\right). \tag{70}$$

Since in practice we typically can only evaluate $p_{\text{target}}$ up to the normalizing constant $\mathcal{Z}$, we can replace it by its unnormalized version $\rho_{\text{target}}$, bringing us unnormalized (discretized) importance

weights, which we call $\widehat{w}$. We can now define the (normalized) effective sampling size (ESS) as

$$\mathrm{ESS} := \frac{\left(\sum_{k=1}^{K} \widehat{w}^{(k)}\right)^2}{K \sum_{k=1}^{K} \left(\widehat{w}^{(k)}\right)^2}, \tag{71}$$

which ranges from 0 to 1, with 1 being optimal. Here and in the sequel $K$ denotes the sample size. Further, we can define the log-variance divergence as

$$D_{\mathrm{LV}} := \mathrm{Var}(\log w) \approx \frac{1}{K-1} \sum_{k=1}^{K} \left(\log \widehat{w}^{(k)} - \frac{1}{K} \sum_{k=1}^{K} \log \widehat{w}^{(k)}\right)^2, \tag{72}$$

whose optimal value is zero. Finally, the normalizing constant $\mathcal{Z}$ can be computed via

$$\mathcal{Z} = \mathbb{E}\left[w\right] \approx \frac{1}{K} \sum_{k=1}^{K} \widehat{w}^{(k)}. \tag{73}$$

### B.3 HYPERPARAMETER DEPENDENCE

In this section, we empirically examine how selected hyperparameters influence the performance of our FTT training algorithm. We focus on the Multiwell problem introduced in Section 4 and consider the setting $d = 10$, $\delta = 2$, with $m \in \{3, 10\}$, corresponding to $2^3$ and $2^{10}$ modes, respectively. We investigate the effect of varying the batch size, the number of basis functions, and the domain shrinking factor $p$ (see Appendix A.4.4), as illustrated in Figure 12 and Figure 13. Unless stated otherwise, we fix the number of basis functions to 10, the batch size to $K = 2^{15}$, and set $p = 0.1$.

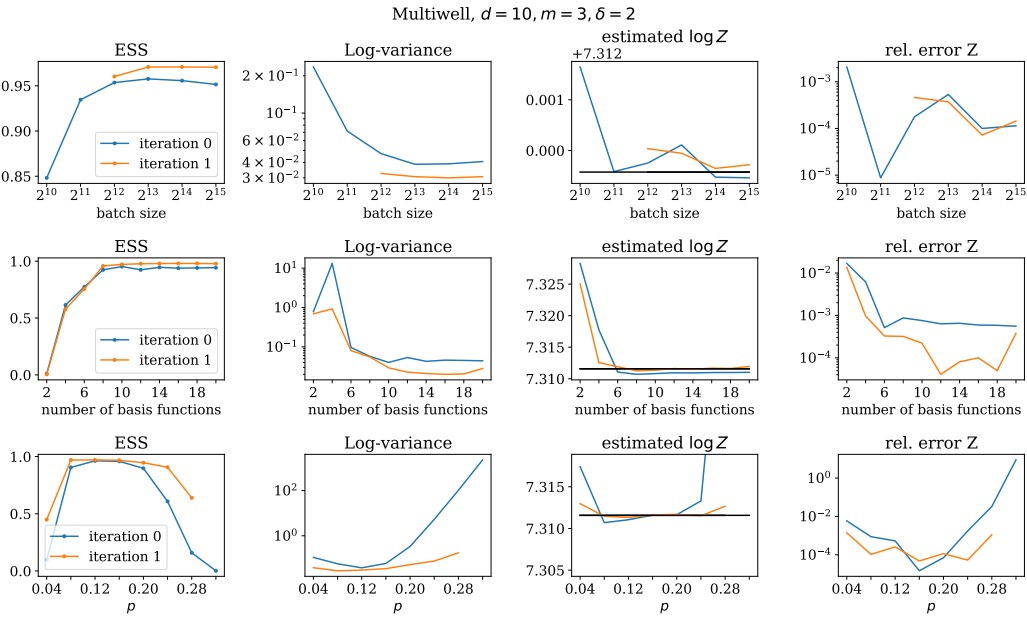

Figure 12: We assess the sensitivity of our algorithm by varying key hyperparameters and recording the effective sample size (ESS), the log-variance divergence, the log-normalizing constant, and the relative error in estimating the normalizing constant. Whenever a value is absent from a plot, the algorithm diverged for that configuration. The results indicate that performance improves with increasing batch size, but the gains tend to saturate beyond a certain threshold. A similar trend is observed for the number of basis functions. In contrast, the parameter $p$ exhibits a clear optimal range: values that are too small or too large degrade performance.

### B.4 COMPARISON WITH ALTERNATIVE SAMPLING METHODS

In Figure 14, we compare the performance of our TTD sampler against various baselines, covering state-of-the-art neural samplers based on ODEs and SDEs and combinations with MCMC methods.

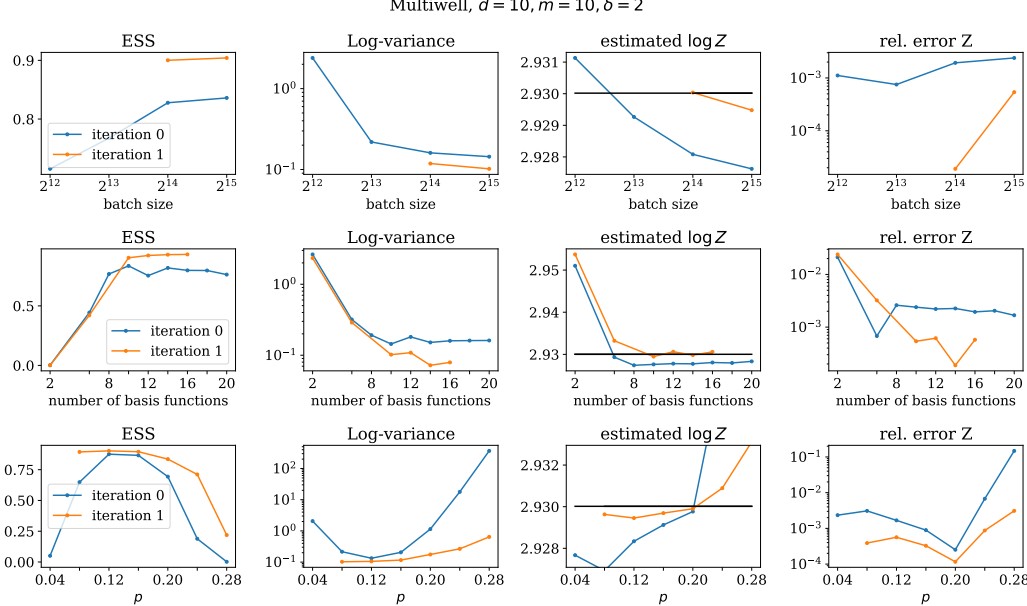

Figure 13: As in Figure 12, we assess the sensitivity of our algorithm by varying key hyperparameters, now on a Multiwell problem with $2^{10}$ modes. We observe the same qualitative trends as before; however, in this highly multimodal setting a larger batch size and a greater number of basis functions are required to stabilize the algorithm.

For all baselines we leverage the public repository by Chen et al. (2024) (which is derived from the repository by Blessing et al. (2024)). We refer to these works for details on each baseline. In particular, we use the default hyperparameters and tune the scale of the prior and diffusion coefficient if applicable. We report the error in estimating $\log Z$ and the ESS (if available for the given baseline) and observe that our TTD sampler consistently achieves better performance for a given time budget as well as better final performance than all considered baselines.

Further, adding to the experiments in Section 4, we additionally compare our TTD sampler with the PIS sampler in more detail (Zhang & Chen, 2022), a well-established diffusion-based sampling method. Note, however, that the underlying PDE is different from our HJB PDE stated in Lemma 2.1. Further, we want to highlight that PIS has a natural advantage when considering the Multiwell problem defined in (19) since its initialization samples from a Gaussian density. In Figure 15 we repeat the same plot as in Figure 3, now integrating PIS as well.

### B.5  ADDITIONAL EXPERIMENT: KITAGAWA NONLINEAR STATE SPACE MODEL

We evaluate our method using a modified version of the Kitagawa nonlinear state space model, designed to isolate high-dimensional geometric complexity while maintaining a unimodal posterior (Kitagawa, 1996). The latent state evolves according to nonlinear dynamics defined by

$$x_n = \frac{1}{2}x_{n-1} + \gamma \frac{x_{n-1}}{1 + x_{n-1}^2} + v_n \tag{74}$$

where $\gamma$ controls the nonlinearity strength and $v_n$ represents Gaussian process noise. Unlike the standard benchmark, we employ a linear observation model to break the symmetry inherent in the original formulation, namely

$$y_n = x_n + w_n. \tag{75}$$

This modification eliminates multi-modality but preserves the challenging correlations in the joint posterior, providing a robust test for sampling efficiency on high-curvature manifolds. Given the fixed initial state $x_0 = 0$ and the observed sequence $y_{1:M}$, the joint log-posterior density $\log \rho(x_{1:M})$ is given by

$$\log \rho_{\text{target}}(x_{1:M} \mid y_{1:M}) = -\sum_{n=1}^{M} \left[ \frac{1}{2\sigma_v^2}\left(x_n - f(x_{n-1})\right)^2 + \frac{1}{2\sigma_w^2}\left(y_n - x_n\right)^2 \right], \tag{76}$$

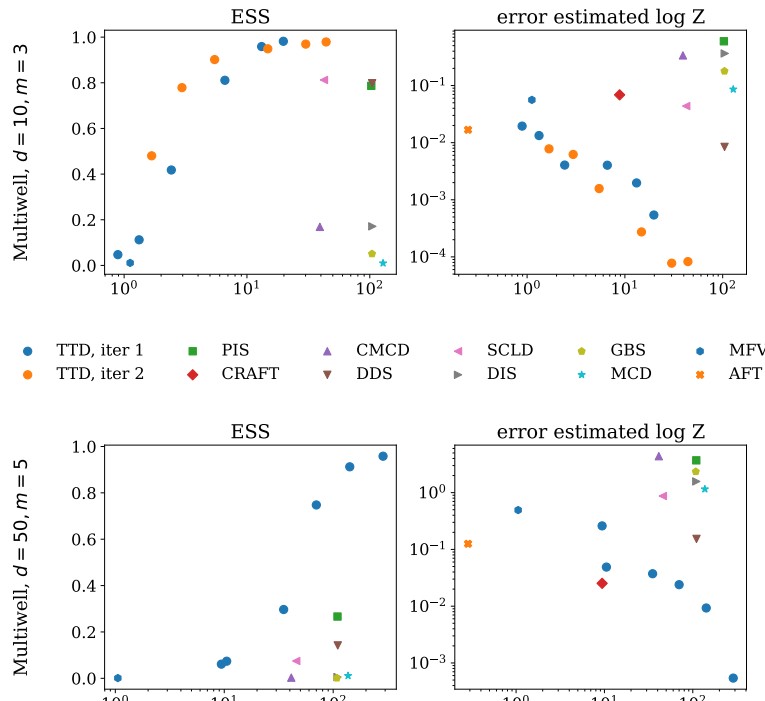

Figure 14: We compare the performance of our TTD sampler against several state-of-the-art baselines, including PIS (Zhang & Chen, 2022), DDS (Vargas et al., 2023), DIS (Richter & Berner, 2024), GBS (Blessing et al., 2024), CMCD (Vargas et al., 2024), SCLD (Chen et al., 2024), CRAFT (Matthews et al., 2022), AFT (Arbel et al., 2021), MFVI (Bishop, 2006), and MCD (Doucet et al., 2022). We see that TTD converges significantly faster and outperforms the baselines in terms of both estimation of normalizing constants as well as ESS (if available for the given baseline). We refer to Appendix B.4 for further details.

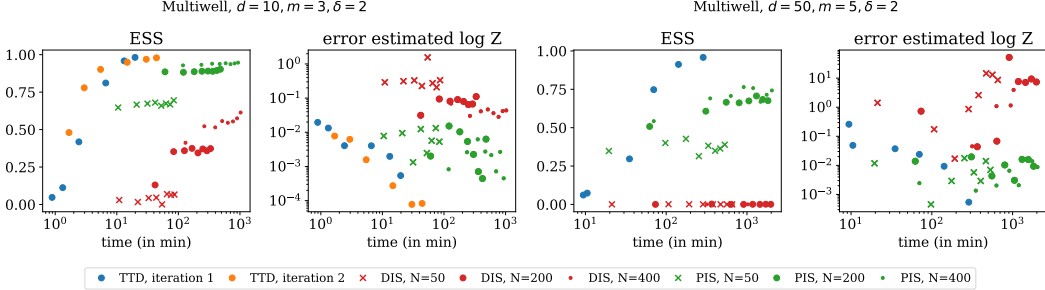

Figure 15: We compare the performance versus runtime of our TTD sampler with DIS (Berner et al., 2024) and PIS Zhang & Chen (2022). By design, our algorithm produces one result per chosen number of steps $N$ (shown as blue and orange dots), whereas DIS and PIS can improve over training time. Accordingly, we evaluate DIS and PIS at equally spaced runtime intervals. In both experiments, our algorithm is not only faster but also achieves better results, particularly in settings where DIS can exhibit instability.

where $f(x_{n-1})$ represents the nonlinear transition mean

$$f(x_{n-1}) = \frac{1}{2}x_{n-1} + \gamma \frac{x_{n-1}}{1 + x_{n-1}^2}. \tag{77}$$

In our experiments we choose $\sigma_v = \sigma_w = 1$ in $d = 10$ and vary the nonlinear strength $\gamma$, see Table 2.

Table 2: We display the effective sample size (ESS) and the log-variance divergence for the Kitagawa nonlinear state space model for varying nonlinear strength $\gamma$ and different amounts of steps $N$.

| $\gamma$ | $N$ | ESS | Log-variance |
|---|---|---|---|
| 0.5 | 2048 | 0.925 | 0.076 |
| | 4096 | 0.937 | 0.068 |
| 1.0 | 2048 | 0.897 | 0.237 |
| | 4096 | 0.911 | 0.133 |
| 1.5 | 2048 | 0.885 | 0.280 |
| | 4096 | 0.898 | 0.192 |
| 2.0 | 2048 | 0.858 | 0.970 |
| | 4096 | 0.871 | 0.612 |

