# OpenReview forum: "Tensor Train Diffusion: A Fast Solver for High-Dimensional Sampling"
_ICLR.cc/2026/Conference — Submitted to ICLR 2026_

### Official Review · Reviewer_Fub6 · 2025-10-26

**Soundness:** 2
**Presentation:** 3
**Contribution:** 2
**Rating:** 2
**Confidence:** 3

**Summary:**

This paper introduces Tensor Train Diffusion (TTD), a novel and efficient numerical solver for diffusion-based sampling. The core challenge in diffusion models is the estimation of the time-dependent score function, which is required to solve the time-reversed SDE. The authors correctly identify that the log-density of this process is governed by a Hamilton-Jacobi-Bellman (HJB) type PDE.

**Strengths:**

1. The paper is easy to follow

2. The computational cost and instability of training samplers are well-known and critical bottlenecks for the widespread use of diffusion models, especially in scientific domains. This paper directly attacks this bottleneck and provides a compelling and effective alternative.

**Weaknesses:**

1. The entire premise of TTD's efficiency hinges on the assumption that the HJB equation's solution $V(x,t)$ possesses a low-rank structure that FTT can exploit The only benchmark used (Multiwell) is quasi-separable by construction  and is therefore an ideal, best-case scenario for tensor-train methods. It is highly questionable whether this low-rank assumption holds for the complex, highly-entangled densities of general datasets (e.g., natural images). If $V(x,t)$ is not low-rank, the required TT ranks $r$ could grow exponentially, making TTD less efficient than NNs. The paper provides no evidence for this more general case and completely omits this crucial discussion of its primary limitation.

2. This work replaces the familiar (though flawed) NN+SGD stack with a set of tools (FTT, ALS, BSDE solvers, HOSVD) that are highly non-trivial and unfamiliar to the vast majority of the NeurIPS community. This creates a significant practical barrier to adoption, reproduction, and extension by other researchers.

3. The main paper presents the method as more straightforward than it is. The appendix reveals that several highly-engineered components are essential for the method to work at all. Most notably, Appendix A.6 states that "a naive choice of $\tau_n$ ... did not yield a convergent scheme", meaning the adaptive regularization scheme is not a minor tweak but a critical, non-obvious part of the algorithm. The same applies to the choice of basis functions (A.3) and the moving approximation domains (A.4.4) .

4. Add a "Limitations and Future Work" section to the main paper. This section must explicitly address Weakness #1. It should state that the method's efficiency relies on the low-rank assumption and discuss which problem classes (like SciML) are a good fit, while acknowledging that generality to highly-entangled data (like images) is an open and challenging question. This makes the paper more rigorous.

5. Briefly allude to the necessity of the adaptive regularization scheme (from A.6) in the main method section (e.g., 3.1 or 4). This gives proper credit to a key part of the algorithm's success without spending too much space.

6. Add a brief ablation or sensitivity analysis in the appendix regarding key "new" hyperparameters, such as the number of basis functions, to aid reproducibility.

**Questions:**

See above

---

> ### Author Response · Authors · 2025-11-21
>
> Dear reviewer Fub6. Thank you very much for your detailed review. We are happy that you appreciate that we attack a notorious bottleneck of existing diffusion-based samplers. Let us address your questions and comments in the sequel.
>
> 1. **Low rank structure of $V$**
>
> We clearly agree with this comment. The efficiency of the proposed solver hinges on the expressivity of the underlying surrogate model. It is not expected that linear approximation classes or non-linear models such as low-rank decompositions, sparse representations and even neural network structures apply in the general case, and hence would solve the sampling problem (up to training time) entirely. With that beeing that, the proposed method relies on the structural properties of the given HJB solution $V(t,\cdot)$, which represents the negative $\log$-density at time $T-t$ of the associated Fokker-Planck equation solution of the backward process. Here, it is up to now clearly an open question, how solutions of the considered HJB behave in the most general case and partial answers have been avaiable only recently [2]. We point out here the fact, that the rank structure for the intermediate densities and the corresponding log-density can also differ. E.g. in the setup of general Gaussian densities, the TT rank can growth exponentially [3], while the log-density has only polynomial growth in $d$ [2]. Solutions of HJB are in fact (negative) log-densities.
>
> From the general perspective, we point out that FTT ranks can be unbounded. As a consequence, the considered xTT format is expressive and suitable for a given function $V(t,\cdot)$ only if the underlying univariate basis functions form good approximations of FTT-component functions and if the associated singular value decay in the FTT decomposition is fast enough, allowing for truncation and hence finite ranks.
>
> Finally, we note that natural images do not fall in our framwork as they do not admit an unnormalized density that can be evaluated. Our work addresses sampling where this unnormalized density is available (such as, e.g., in the natural sciences or Bayesian statistics).
>
>
>
> 2. **Used tools are non-trivial and unfamiliar to machine learning community**
>
> We are glad that you agree that neural networks (NNs) and stochastic gradient descent (SGD) have notable limitations. This is precisely our motivation for introducing alternative and novel tools for sampling from unnormalized densities. Only with these tools are we able to achieve our main contribution: a significant improvement in computational efficiency while maintaining high simulation performance (see Figure 3, where we compare to methods that rely on NNs and SGD). While we acknowledge that the methods we use may not be immediately intuitive, we have made efforts to introduce them carefully (see also the added explanations highlighted in blue). We also note that tensor trains (TTs) have previously gained recognition in the machine learning community, including a Best Paper Award at ICML 2021 (see [1]); multiple other TT-based submissions have appeared in ICLR 2026 as well. Furthermore, we plan to release our code upon acceptance, enabling others to use our methods without requiring full mastery of all mathematical details.
>
> We revisited section 3 (*Tensor trains as approximating functions*) to motivate more intuition to the low-rank format for readers non-familiar with the non-linear approximation class of (extended) tensor trains and refer to the extended Appendix A for more details, e.g. the evaluation of such functions.
>
> We would greatly appreciate any suggestions on where we could further improve our explanations, as such feedback would be extremely valuable to us.
>
>
> 3. **In the main part, the method is presented as more straightforward as it is**
>
> Thank you for this important comment. We agree that several technical developments were necessary to make our method perform as well as it does, and we consider these developments to be essential contributions of our work. We have already highlighted this in the main text and linked to the corresponding details in the appendix, for example, Remark 3.1 (evaluations outside the trajectory domain). However, we agree that these details deserve more prominence in the main text, and we have accordingly added further explanations, see the parts marked in blue, in particular Remark 3.3. Please let us know if you find this revision more suitable; otherwise, we would be happy to further improve the presentation, as the paper greatly benefits from your constructive comments.

---

> > ### Author Response · Authors · 2025-11-21
> >
> > 4. **Add "Limitations and Future Work" section**
> >
> > Thank you for this suggestion; we have added a "Limitations" section in the new Section 4.1. Please let us know if you feel that any details are still missing. An outlook on future work is already provided in Section 5 (with even more details now).
> >
> > 5. **Allude to the necessity of the adaptive regularization scheme in the main method**
> >
> > Thank you for this comment. As mentioned before, we have now highlighted this aspect more. For space contraints, we need to keep the details in the appendix. Please let us know if you are missing further information or if we should place it more prominently.
> >
> > 6. **Sensitivity analysis in the appendix regarding key hyperparameters**
> >
> > Thank you very much for this valuable suggestion. We have added Section B.2 to the appendix to investigate the dependence on hyperparameters; see in particular the new Figures 8 and 9. These results indicate that the algorithm remains reasonably stable across a broad range of sensible hyperparameter choices, but - as expected - becomes less stable under extreme or unreasonable settings. Please let us know if you would be interested in additional experiments.
> >
> >
> >
> >
> > [1] Richter, L., Sallandt, L., & Nüsken, N. (2021, July). Solving high-dimensional parabolic PDEs using the tensor train format. In International Conference on Machine Learning (pp. 8998-9009). PMLR.
> >
> >
> > [2] Gruhlke, R. and Sommer, D. and Kirstein, M. and Eigel, M. and  Schillings, C.  Reverse Diffusion Sampling with Tensor Train approximations of Hamilton-Jacobi-Bellmann equations.(2025) SIAM Journal on Scientific Computing.
> >
> > [3] Rohrbach, P. B. and Dolgov, S. and Grasedyck, L. and Scheichl, R. Rank bounds for approximating gaussian densities in the tensor-train format. (2022) SIAM/ASA Journal on Uncertainty Quantification

---

> ### Author Response · Authors · 2025-12-03
> **General response**
>
> We thank the reviewer again for their insightful and constructive comments. We have addressed these points in our updated manuscript and provided a high-level overview in our general response, 'Summary of contributions and revisions.'

---

### Official Review · Reviewer_ZdeQ · 2025-11-01

**Soundness:** 3
**Presentation:** 3
**Contribution:** 3
**Rating:** 6
**Confidence:** 2

**Summary:**

This paper introduces Tensor Train Diffusion (TTD), a novel method for sampling from unnormalized high-dimensional probability distributions. The approach solves the Hamilton-Jacobi-Bellman (HJB) PDE underlying diffusion-based sampling using functional tensor train (FTT) representations rather than neural networks. The key innovation is combining the FTT format—which efficiently represents high-dimensional functions by exploiting low-rank structure.

**Strengths:**

The paper makes a novel and well-motivated methodological contribution by applying tensor train representations to solve HJB equations for diffusion-based sampling, directly addressing real limitations of neural network-based methods such as long training times and hyperparameter sensitivity. The theoretical foundation is strong, with clear connections established between diffusion models, HJB PDEs, and BSDEs, and rigorous derivations from the BSDE formulation to the discrete loss function.

**Weaknesses:**

The experimental validation is limited, focusing exclusively on the Multiwell benchmark family, and would be significantly strengthened by demonstrating performance on other problem classes.

**Questions:**

How does the required TT rank scale with dimension $d$, mode separation $\delta$, and number of modes $m$? Can you characterize this relationship theoretically or empirically?

---

> ### Author Response · Authors · 2025-11-21
>
> Dear Reviewer ZdeQ. We thank you very much for your review. We are happy that you appreciate that we replace neural networks by tensor trains in diffusion-based sampling and that you in particular value our theoretical contibution as strong. Let us address your comments and questions in the sequel.
>
> * **Limited experimental validation**
>
> Thank you for this comment. First, note the Multiwell is a well-established benchmark problem in the (diffusion-based) sampling literature, resembling molecular dynamics problems and providing an indication of performance in practical applications. At the same time, it offers a controlled setting for systematically studying the three key challenges in sampling: (1) the dimensionality $d$, (2) the mode separation $\delta$, and (3) the number of modes $2^m$. This is why we focused on these class of problems. We will, however, try to add new experiments in the short time of the rebuttal and will give you another update soon.
>
>
> * **How does the required TT rank scale with dimension $d$, mode separation $\delta$, and number of modes $m$? Can you characterize this relationship theoretically or empirically?**
>
> Thank you for the question. The principal motivation for using low-rank formats lies in their favorable scaling properties. When representing a function in the xTT format with a given rank vector $\mathbf{r} = (r_1, \ldots, r_{d-1})$, the number of degrees of freedom scales as $\mathcal{O}(dM \max\limits_{i=0,\ldots,d-1}r_ir_{i+1})$, with the convention $r_0 = r_d = 1$ and $M$ being the maximum mode size, i.e. the number of univariate basis functions in each coordinate. This scaling is linear in the dimension $d$. While the rank behavior itself depends on the specific problem, our experiments indicate that the ranks remained stable across the investigated mode separations $\delta$ and numbers of modes $m$.

---

> > ### Author Response · Authors · 2025-12-03
> > **General response**
> >
> > We thank the reviewer again for their insightful and constructive comments. We have addressed these points in our updated manuscript and provided a high-level overview in our general response, 'Summary of contributions and revisions.'

---

### Official Review · Reviewer_tHHw · 2025-11-01

**Soundness:** 3
**Presentation:** 3
**Contribution:** 3
**Rating:** 6
**Confidence:** 2

**Summary:**

The paper proposes Tensor Train Diffusion (TTD), a solver for the HJB-type PDE that underlies diffusion-based sampling from complex, high-dimensional unnormalized densities. Instead of training a large neural PDE surrogate, TTD represents the value/log-density function with a functional tensor train (FTT) and derives a backward-in-time scheme from the HJB–BSDE connection. This yields a regression objective based on a BSDE residual, and the FTT parameters are optimized by alternating least squares (ALS) sweeps.

**Strengths:**

1. It replaces large neural surrogates with a functional tensor-train (FTT) representation, which avoids costly target-density evaluations and reliance on higher-order automatic differentiations.

2. Each TT core is updated by solving a regularized linear least-squares subproblem, enabling stable, fast sweeps.

3. On 10D and 50D multi-modal targets, runtime scales linearly with steps, and TTD shows favorable ESS/log-Z behavior versus DIS; an additional plot also compares against PIS.

**Weaknesses:**

1. The core experimental results are on Multiwell only (two setups: 10D/50D).

**Questions:**

1. How sensitive are results to the rank-adaptation threshold and update frequency?

2. Under what structural conditions on V(x,t) do you expect a low TT rank?

---

> ### Author Response · Authors · 2025-11-21
>
> Dear Reviewer tHHw. We thank you very much for your careful review. We are happy that you appreciate the clear advantages of our novel method, i.e. less target evaluation and significantly faster training. Let us address your questions and comments in the sequel.
>
> * **More experiments**
>
> Thank you for this comment. First, note the Multiwell is a well-established benchmark problem in the (diffusion-based) sampling literature, resembling molecular dynamics problems and providing an indication of performance in practical applications. At the same time, it offers a controlled setting for systematically studying the three key challenges in sampling: (1) the dimensionality $d$, (2) the mode separation $\delta$, and (3) the number of modes $2^m$. This is why we focused on these class of problems. We will, however, try to add new experiments in the short time of the rebuttal and will give you another update soon.
>
> * **How sensitive are results to the rank-adaptation threshold and update frequency?**
>
> The rank-adaption treshold is a critical hyperparameter in the regression scheme as it directly influences the approximation quality of the xTT approximation model. First note, that the approximation quality is limited by the underlying linear space of tensorized univariate basis functions. Within this range of maximally possible approximation power, the non-linearity of the approximation model is introduced by the rank. If the latter is choosen too small, the approximation quality may deviate from the best possible by magnitutes. If the rank is choosen too big, while the approximation quality is given, the efficiency of the surrogate class and its training is limited. Hence the choice of treshold balances between approximation quality and efficiency.
>
> In the ideal setting, an update frequency is set to $1$. However, when looking for classical solutions, the HJB solution $t\mapsto V(t,\cdot)$ is expected to change continuously, hence singular values in the FTT represenation are expected to change continuously. With that beeing said, the update frequency can be coarsened depending on the the underlying time discretization to reduce the learning costs. In our experiments we used frequency of $5$-$10$.
>
>
> * **Under what structural conditions on V(x,t) do you expect a low TT rank?**
>
> This is a very interesting question which touches open questions in the solution structure of the associated HJB equation and we like to  briefly discuss the current state of the art.
>
> While first analysis for particular instances of rank behaviour of the considered HJB equation were analyzed in [4], the complete answer in the general case is unknown. The HJB has two relevant ingredients that manipulate the rank structure between the boundary conditions with possible accessable rank, i.e. the log-target density and the log-prior density. The non-linear term $\tfrac{1}{2}\|\sigma^\top \nabla \cdot\|^2$ is a main contribution for possible rank growth, while the diffusion related terms that pull the solution to the log-density of the stationary distribution should reduce the rank.
>
> A quite detailled analysis of rank behaviour for certain regularity classes (Besov-, Sobolev spaces) was derived and presented in [1,2,3]. However note, that for any $t$ the solution $V(t,\cdot)$ may not be a function in such regularity classes, since we are working on the unbounded domain $\mathbb{R}^d$; e.g. the log-Gaussian $\tfrac{1}{2}\sum\limits_{i=1}^d x_i^2$ is not even in $L^2(\mathbb{R}^d)$.
>
> Hence, the theory does not directly apply. One approach is to  extend the latter theory to weighted regularity spaces, being subspaces of $L^2(\mathbb{R}^d, \rho_t)$, where $\rho_t\sim \exp(-V(t,\cdot))$. This is a topic of recent research and out of the scope of the submitted contribution. In our work, we use the concept of truncation of the computational domain, again leading (locally) to the possible application of analysis in the spaces considered in [1,2,3]. However, the regularity/integrability of HJB solutions $t\mapsto V(t,\cdot)$ (which is a non-linear parabolic PDE) up to the knowledge of the authors is unknown, making application of the theory not as straightforward.
>
> As a consequence, with our contribution we hope to shed some light empirically, when it comes to understanding the mechanics  of HJB equations and structure of its solutions. We added a related discussion in the *Conclusion & Discussion* section.
>
> [1] Ali, M. and Nouy, A. Approximation with tensor networks. Part II: Approximation rates for smoothness classes. (2020).
>
> [2] Griebel, M. and Harbrecht, H. and Schneider, R. (2023). Mathematics of Computation.
>
> [3] Bachmayr, M. Low-rank tensor methods for partial differential equations. (2023). Acta Numerica.
>
> [4] Gruhlke, R. and Sommer, D. and Kirstein, M. and Eigel, M. and  Schillings, C. (2025) Reverse Diffusion Sampling with Tensor Train approximations of Hamilton-Jacobi-Bellmann equations. SIAM Journal on Scientific Computing.

---

> > ### Author Response · Authors · 2025-12-03
> > **General response**
> >
> > We thank the reviewer again for their insightful and constructive comments. We have addressed these points in our updated manuscript and provided a high-level overview in our general response, 'Summary of contributions and revisions.'

---

### Official Review · Reviewer_RsEA · 2025-11-02

**Soundness:** 2
**Presentation:** 2
**Contribution:** 2
**Rating:** 2
**Confidence:** 4

**Summary:**

This paper proposes tensor train diffusion, a method for sampling from unnormalized probability densities, combining tensor train representations with SDEs. The authors frame their approach as a diffusion model innovation, positioning it within the recent generative modeling literature. The method employs tensor train decompositions for function approximation and solves Hamilton-Jacobi-Bellman equations through backward-in-time iterations. Numerical experiments on a multiwell synthetic function (with d=10 and d=50) are used to compare against two approaches that also leverage SDE frameworks for addressing this problem.

**Strengths:**

The paper addresses a fundamental problem in computational statistics and scientific computing that has significant practical importance across many domains from Bayesian inference to statistical physics. The methodology is documented in considerable detail in the main paper and the appendices. A detailed mathematical exposition of the tensor train decomposition framework, the BSDE formulation, and the connection to Hamilton-Jacobi-Bellman equations are provided.

**Weaknesses:**

The introduction conflates two fundamentally distinct problems by citing diffusion model literature (Ho et al., 2020; Song et al., 2021) as motivation for sampling from known unnormalized densities. This is a classical problem dating back to Metropolis (1953) and has been extensively studied in various fields since then. The papers by Ho et al. and Song et al. address a distinct problem: learning to generate samples from an unknown distribution given only empirical samples, without access to the functional form of the density. Presenting the classical problem of generating samples from unnormalized densities as a novel application of diffusion models misrepresents the problem's rich heritage.

The authors cite Dai et al. (2022) to support their claim that MCMC methods “typically require substantial tuning and long runtimes.” However, this citation is not appropriate as Dai et al.’s paper titled “An Invitation to Sequential Monte Carlo Samplers” advocates for wider adoption of sequential Monte Carlo methods in statistics emphasizing their ability to leverage parallel processing resources among other potential benefits. This reference actually undermines rather than supports the authors’ critique of Monte Carlo methods.

If I understand correctly, the proposed method appears to rely on annealed Langevin dynamics for initialization, as shown in Figure 1 where Langevin already discovers both modes. Given that the paper dismisses classical methods as inefficient, I find it somewhat contradictory that TTD requires a classical MCMC method to provide a near-solution before it can even begin? How would TTD perform without this classical initialization, and doesn't this dependency suggest that classical methods are actually doing most of the work? It is not clear to me if the baseline methods in the numerical studies were also allowed the benefit of this initialization strategy or if they started from scratch.

The methodology writeup begins with an SDE in equation (2) that maps samples from the target distribution $p_{target}$ to samples from a known reference/prior distribution. However, in the problem setting where $\rho_{target}$ is known and samples from $p_{target}$ are not provided, this forward process cannot actually be simulated. This forward SDE appears to serve no computational purpose in the subsequent algorithm and seems to exist solely to maintain alignment with the diffusion model framework of Ho et al. and Song et al. that addresses a different problem statement.

The paper appears to force-fit the diffusion model-based generative modeling framework of Song et al. The setting in Song et al. involving forward and time-reversed SDEs makes sense when we have access to samples drawn from the unknown target distribution. Here, since $\rho_{target}$ is given, the score function of the target distribution is also known. The elaborate construction through HJB equations and BSDEs essentially reduces to solving for a transport map that pushes samples from a known reference distribution to samples from the target distribution. Therefore, this problem can be approached more directly without the detours that the paper takes.

I have several questions and concerns about the numerical studies - please see next section.

**Questions:**

- Given that the proposed method essentially applies tensor train decomposition to a transport problem, what specific advantages does the diffusion model framing provide over directly solving the optimal transport or Schrödinger bridge problem?
- Could the authors clarify the distinction between their "extended Tensor Train (xTT)" formulation and the standard functional tensor train (FTT) construction? Unless I am mistaken, the proposed construction appears identical to the  FTT format (Oseledets, 2013) that is widely used in the literature. If the proposed xTT introduces novel aspects beyond the established FTT framework, it would be helpful to explicitly highlight these contributions.
- The evaluation focuses solely on a synthetic multiwell test function with d=10 and d=50. Why aren't standard benchmarks from the literature included, such as those used in Zhang & Chen (2022), or the extensive test suites from the normalizing flow literature?
- The experiments compare only against DIS (Berner et al., 2024) and PIS (Zhang & Chen, 2022). Why are there no comparisons against well-established workhorses such as NUTS, HMC, pSMC, which have been developed to exactly address this problem class?
- The proposed method includes an initial regression of $\log \rho_{target}$ as mentioned in Remark 2.5. How sensitive is the overall algorithm performance to the quality of this initial regression?
- Alternating least squares is known to suffer from slow convergence. How do the authors overcome this limitation in practice, and have they tested the method on problems beyond the relatively simple multiwell distributions? Was the initialization step motivated by the  poor convergence of ALS?
- Do all methods use the same initialization strategy? Specifically, does TTD's use of annealed Langevin initialization (which Figure 1 shows already discovers all modes) provide an unfair advantage over DIS and PIS, which appear to start from scratch? For fair comparison, either all methods should start from the same Langevin-initialized point, or all should start from a simple reference distribution.
- Could the authors provide detailed specifications of the hyperparameters and implementation choices used for the baseline methods, particularly DIS? What do the N values (50, 200, 400) represent for DIS? Additionally, were the hyperparameters for DIS tuned for these specific problems, or were default values from the original paper used? Given that Berner et al. report strong performance on similar problems with different settings, understanding whether the baselines received comparable optimization effort to the proposed method would help readers assess the fairness of the comparisons.
- Could the authors provide a detailed computational cost breakdown, specifically the number of evaluations of $\rho_{target}$  versus other computational steps? The plots show runtimes of up to 300 minutes for "two iterations", which I am unable to make sense of.  A breakdown showing number of evaluations of $\rho_{target}$, time spent in the initial regression/preprocessing (Remark 2.5), ALS iteration counts and costs, etc would help readers understand where the computational budget is being spent and whether the method would remain competitive for problems where density evaluation is expensive (e.g., problems requiring numerical integration or simulation).

---

> ### Author Response · Authors · 2025-11-21
>
> Dear Reviewer RsEA. Thank you very much for your detailed review. We greatly appreciate your careful reading and are glad to have received many concrete suggestions. We are happy that you value our problem as significant and appreciate that we introcuce our method in detail and with mathematical rigor. We address your comments in the following.
>
> * **Traditional diffusion models vs. diffusion-based sampling.**
>
> Thank you for raising this point. We believe there may be a misunderstanding. While diffusion models were indeed originally introduced for the setting in which samples from the target distribution are available but the density is unknown, the inverse problem - where the density is known but samples are not (i.e., the sampling problem) - has been studied extensively over the past three years. This setting, often referred to as *diffusion-based sampling*, builds upon the same theoretical foundations as diffusion models. It was first introduced in [1, 2] and has since been further developed in numerous follow-up works (see, e.g., [3, 4]). All of these approaches rely on the time-reversal of stochastic differential equations (SDEs), analogous to diffusion models. In contrast to classical sampling algorithms (such as, e.g., MCMC), diffusion-based sampling methods aim to converge to the target distribution within finite time. We did not intend to present the application of diffusion models to sampling as a novel contribution and, in fact, explicitly reference several prior works on diffusion-based sampling throughout our paper.
>
> * **Dai et al. (2022) not a good reference for MCMC challenges.**
>
> Thank you for this careful observation. We agree that Dai et al. (2022) is not the most suitable reference for illustrating the well-known issue that MCMC-type algorithms struggle to move between distant high-probability regions due to the energy barriers separating them. We have therefore replaced this citation with the recent work [5], which provides a detailed analysis of this phenomenon, as well as the standard textbook [6], which discusses poor mixing (i.e., slow convergence) in complex state spaces, including multimodal distributions (highlighted in blue in the PDF).
>
> * **Annealed Langevin initialization and necessity to find all modes**
>
> This is an important point, and we believe there may be a small misunderstanding. In Figure 1, our intention is to illustrate that the initialization procedure (e.g., annealed Langevin dynamics, though standard Langevin would also suffice) does not need to discover all modes of the target distribution. The key mechanism of TTD is the outer loop (see Algorithm 1): in the first iteration, the optimal control is approximated along trajectories that may not yet explore all modes. In subsequent outer iterations, however, the learned control can guide the process toward additional modes that were previously missed. In our experiments, we observed that this outer loop typically converges. We have added a clarification regarding this behavior in the newly introduced Section 4.1 ("Limitations"). Furthermore, the baseline methods are also effectively initialized using Langevin dynamics (see, e.g., Eqs. (24) and (141) in [1]), so we consider our comparison to be fair.
>
> * **Forward procees from target to prior appears to serve no computational purpose**
>
> Thank you for raising this important point. You are correct that, unlike in classical diffusion models, the forward process from the target distribution to the prior cannot be simulated, as we do not have access to samples from the target. However, this SDE, denoted as (2), is still essential: it enables the derivation of the corresponding PDE (4) and forms the basis for our backward algorithm via BSDE theory. This connection is explained in detail in Section 2.2 and summarized in Algorithm 1. Importantly, such a backward algorithm is only feasible in the setting where the (unnormalized) target density is available - hence it is not applicable in the classical diffusion model setting. What we simulate in practice is the "denoising SDE" (3), which incorporates the currently learned control $u$ and transports samples from the prior toward the target distribution. This mechanism is directly analogous to the sampling procedure in diffusion models. In data-free diffusion-based sampling, this setup is standard (cf. [1, 2, 3, 4]). We have added two additional sentences at the beginning of Section 2.1 to make this clearer. Please let us know if any further clarification would be helpful, as this is a central conceptual point.

---

> > ### Author Response · Authors · 2025-11-21
> >
> > * **The problem of dynamical measure transport can be approached more directly without the detours that the paper takes.**
> >
> > Connecting to the arguments provided in our previous responses, we respectfully disagree with this point. In our understanding, the reasoning used to derive our backward-in-time algorithm is both necessary and aligned with the now-standard methodology in diffusion-based sampling (see, e.g., [1, 2, 3, 4]). If you have further questions or suggestions on how the derivation could be shortened or simplified, we would be grateful to hear them. We would also like to emphasize that, to the best of our knowledge, our work is the first to propose a backward-in-time algorithm in the context of diffusion-based sampling.
> >
> > ---
> >
> > Let us now answer your questions in the sequel.
> >
> > > Given that the proposed method essentially applies tensor train decomposition to a transport problem, what specific advantages does the diffusion model framing provide over directly solving the optimal transport or Schrödinger bridge problem?
> >
> > Thank you for this excellent question. Our approach is based on two synergistic advantages. **Why FTTs:** To our knowledge, FTTs are novel for diffusion-based sampling problems. Unlike NNs, they exploit low-rank structure, which enables efficient regression-based solvers in the training stage. This is the key to the substantially faster optimization we demonstrate in Figure 3 (note the log scale) and aligns with similar findings in [7]. **Why time-reversed diffusion**: The diffusion framework provides a natural BSDE viewpoint. This is crucial, as it allows us to derive a backward-in-time algorithm specifically structured to leverage the fast, regression-based solvers that FTTs provide. In short, the BSDE framework provides the algorithmic structure and FTTs provide the efficient computational representation. It remains an open question whether a direct optimal transport formulation would provide an equally convenient structure for this type of fast, regression-based optimization.
> >
> > > Could the authors clarify the distinction between their "extended Tensor Train (xTT)" formulation and the standard functional tensor train (FTT) construction? Unless I am mistaken, the proposed construction appears identical to the FTT format (Oseledets, 2013) that is widely used in the literature. If the proposed xTT introduces novel aspects beyond the established FTT framework, it would be helpful to explicitly highlight these contributions.
> >
> > Thank you for raising this point - it touches on an important conceptual distinction between several related low-rank function representations. The functional tensor train (FTT) format, as introduced in Oseledets (2013), is a separability concept defined directly at the level of functions, it can be unbounded. In that sense, the FTT format is strictly more general and certainly encompasses the representation we refer to as xTT in this work.
> >
> > At the opposite end of the spectrum lies the standard tensor train (TT) format, which is defined purely at an algebraic/discrete level. The extended tensor train formulation ([8], abbreviated as xTT in this work and ETT in [11]), sometimes refered as Extended Functional Tensor Train (EFTT, [9]) or spectral tensor-train decomposition [10], used in this work, occupies an intermediate position. In xTT, each TT core is augmented with a fixed univariate basis (or feature map) that contracts with the non-rank degrees of freedom. Thus, xTT can be viewed in two equivalent ways:
> >
> > From the TT perspective: xTT is an extension of the standard TT format, enriching each core with an internal functional structure via basis functions.
> >
> > From the FTT perspective: xTT corresponds to a discretization (or parametrization) of an FTT representation through specific choices of finitely many univariate bases/feature functions and hence the rank is finite.
> >
> > With that being said, an extended (functional) tensor train format relies on the choice of underlying basis/feature map design. While the FTT theory establishes existence results and provides examples of functions with finite FTT rank, the extended (functional) tensor train construction specifies how these function factors are realized in practice through basis functions, enabling concrete numerical algorithms such as regression and implementations. This modeling choice is essential for the computational methods developed in the paper.
> >
> > We added a related discussion and references in the related work section and in Section 3, when introducing the format.

---

> ### Author Response · Authors · 2025-11-21
>
> > The evaluation focuses solely on a synthetic multiwell test function with d=10 and d=50. Why aren't standard benchmarks from the literature included, such as those used in Zhang & Chen (2022), or the extensive test suites from the normalizing flow literature?
>
> Thank you for this comment. First, note the Multiwell is a well-established benchmark problem in the (diffusion-based) sampling literature, resembling molecular dynamics problems and providing an indication of performance in practical applications. At the same time, it offers a controlled setting for systematically studying the three key challenges in sampling: (1) the dimensionality $d$, (2) the mode separation $\delta$, and (3) the number of modes $2^m$. This is why we focused on this class of problems. We will, however, try to add new experiments in the short time of the rebuttal and will give you another update soon.
>
> > The experiments compare only against DIS (Berner et al., 2024) and PIS (Zhang & Chen, 2022). Why are there no comparisons against well-established workhorses such as NUTS, HMC, pSMC, which have been developed to exactly address this problem class?
>
> Thank you for the comment. We will work on providing additional comparisons. However, we note that we have already compared our method against PIS and DIS, which are now established diffusion-based samplers and have themselves outperformed multiple standard sampling algorithms (see [1, 12]).
>
> > The proposed method includes an initial regression of $\log \rho_\mathrm{target}$ as mentioned in Remark 2.5. How sensitive is the overall algorithm performance to the quality of this initial regression?
>
> Thank you for this question. We use the regression of $\log \rho_{target}$ via xTTs solely to initialize $V_{N-1}$, which is subsequently optimized using the BSDE-based loss in (9). Therefore, an imperfect initial regression is not expected to be problematic, as long as it provides a better starting point than leaving $V_{N-1}$ uninitialized.

---

> > ### Author Response · Authors · 2025-11-21
> >
> > > Alternating least squares is known to suffer from slow convergence. How do the authors overcome this limitation in practice, and have they tested the method on problems beyond the relatively simple multiwell distributions? Was the initialization step motivated by the poor convergence of ALS?
> >
> > Thank you for this insightful comment. We agree that classical ALS can exhibit slow convergence, particularly when the tensor cores are poorly initialized or when the underlying problem is ill-conditioned. In our setting, however, this issue is mitigated by a combination of problem structure, tailored initialization, and algorithmic safeguards, as detailed below.
> >
> > **Structured objective and well-conditioned subproblems.** The loss functions arising from our formulation lead to subproblems that are comparatively well-conditioned. Empirically, this results in substantially faster convergence than the worst-case scenarios typically associated with generic ALS. In all our experiments, each core update converges in only a few inner iterations.
> >
> > **Initialization strategy.** The initialization procedure is motivated by the structure of the explicit optimization loss used in this work:
> >
> > (i) The loss depends on a step size. When this step size approaches zero, the loss converges to a standard $L^2$-loss. Hence, initializing via an $L^2$-minimization problem - which is better understood in terms of conditioning and sample complexity - provides a reliable proxy for the target problem. Because the overall optimization is nonlinear and nonconvex, high-quality initialization is crucial for reaching a good basin of attraction.
> >
> > (ii) Evaluating the $L^2$-loss is computationally cheaper, since it only requires function evaluations and avoids repeated differentiation of the model. This makes it an efficient first step.
> >
> > (iii) The explicit loss used in our method is empirically less robust to naive initialization than the $L^2$-based approach. The $L^2$-initialization therefore stabilizes the subsequent ALS iterations and accelerates convergence, rather than compensating for a fundamental deficiency of ALS.
> >
> > **Regularization and rank control.** ALS convergence in our setting benefits from adaptive regularization at each time step. Because the univariate basis is
> > $H_{mix}^2$-orthonormal, Tikhonov regularization in each algebraic microstep corresponds to regularization in the functional $H_{mix}^2$-norm by Plancherel. The appropriate regularization strength changes as the target evolves from ⁡log-density to log-prior (on truncated domains), making an adaptive choice essential for robustness. We present an related discussion in the Appendix A.6 (*Choice of regularization magnitude $\tau_n$*), so please let us know if any further clarification would be helpful. Moreover, alternating schemes such as MALS or DMRG naturally support rank adaptivity. Since the optimization is performed over a sequence of time steps with smoothly varying targets, the TT solution obtained at the previous step serves as an excellent initialization for the next one. As a result, each ALS call within the time-stepping loop requires only a small number of sweeps.
> >
> > Note that our rank adaptive scheme is seperate from the choice of loss optimization described in Algorithm 4 (*Adaptive Rank Adjustment for Tensor Train Learning*). With that beeing said, alternatively so alternating minimization schemes such as the presented ALS, we can also incoorperate Alternating Steepest Descent (ASD) or even Riemannian optimization based on automatic differentiation enabled by the fact that Tensor Trains with fixed rank form a Riemannian manifold. We added a new section *A.7 Beyond ALS solvers for minimization* to discuss related things.
> >
> > As mentioned before, we will try to add more experiments soon.
> >
> > > Do all methods use the same initialization strategy? Specifically, does TTD's use of annealed Langevin initialization (which Figure 1 shows already discovers all modes) provide an unfair advantage over DIS and PIS, which appear to start from scratch? For fair comparison, either all methods should start from the same Langevin-initialized point, or all should start from a simple reference distribution.
> >
> > We have already answered this question in the comments above. Please let us know if you have further questions.

---

> ### Author Response · Authors · 2025-11-21
>
> > Could the authors provide detailed specifications of the hyperparameters and implementation choices used for the baseline methods, particularly DIS? What do the N values (50, 200, 400) represent for DIS? Additionally, were the hyperparameters for DIS tuned for these specific problems, or were default values from the original paper used? Given that Berner et al. report strong performance on similar problems with different settings, understanding whether the baselines received comparable optimization effort to the proposed method would help readers assess the fairness of the comparisons.
>
> To provide fair comparisons, we used problems that have also been considered in the paper that developed DIS [1,3]. Thus, we could use their optimized hyperparameters and reproduce their results using the official repository. The number $N$ represents the number of time-steps used by DIS and PIS to simulate the SDE with the Euler-Maruyama scheme during training.
>
>
> > Could the authors provide a detailed computational cost breakdown, specifically the number of evaluations of $\rho_\mathrm{target}$ versus other computational steps? The plots show runtimes of up to 300 minutes for "two iterations", which I am unable to make sense of. A breakdown showing number of evaluations of $\rho_\mathrm{target}$, time spent in the initial regression/preprocessing (Remark 2.5), ALS iteration counts and costs, etc would help readers understand where the computational budget is being spent and whether the method would remain competitive for problems where density evaluation is expensive (e.g., problems requiring numerical integration or simulation).
>
> Thank you for this excellent question. In the numerical experiment discussed in Section 4, we only consider target evaluations for the initial data coming from the annealed Langevin run and for the evaluation at the terminal time. These are used to perform the initial fitting of $\rho_\mathrm{target}$ (see Remark 3.2) and to initialize $V_N$. During the course of the optimization (i.e., learning the optimal control $u^*$), no further target evaluations are necessary, which is why we cannot provide the plots you requested. This behavior is fundamentally different from previous diffusion-based sampling approaches using neural networks, where the transport SDE (3) must be repeatedly simulated with the current $u$ and the target repeatedly evaluated in the loss, as those methods rely on stochastic gradient descent. To give an example, in our $50$-dimensional Multiwell experiment, we considered $10000 \times N$ target evaluations for the annealed Langevin simulation and $50000$ additional evaluations for fitting $\rho_\mathrm{target}$. This is significantly fewer than typically required in diffusion-based sampling; e.g., in DIS and PIS, the number of target evaluations can be around $10^9$ to $10^{11}$ and even more. We note that an uninformed first SDE could also be used instead of annealed Langevin, which would further reduce the number of target evaluations, but this could also result in less informative initial trajectories. Furthermore, we could also use the Functional Tensor Train approximation of the target density for the annealed Langevin simulation, which would further substantially reduce the number of target evaluations. We refer the reviewer to Remark 2.5.

---

> > ### Author Response · Authors · 2025-11-21
> >
> > [1] Berner, Julius, Lorenz Richter, and Karen Ullrich. "An optimal control perspective on diffusion-based generative modeling." arXiv preprint arXiv:2211.01364 (2022).
> >
> > [2] Vargas, Francisco, Will Grathwohl, and Arnaud Doucet. "Denoising diffusion samplers." arXiv preprint arXiv:2302.13834 (2023).
> >
> > [3] Richter, L., & Berner, J. (2023). Improved sampling via learned diffusions. arXiv preprint arXiv:2307.01198.
> >
> > [4] Vargas, F., Padhy, S., Blessing, D., & Nüsken, N. (2023). Transport meets variational inference: Controlled monte carlo diffusions. arXiv preprint arXiv:2307.01050.
> >
> > [5] Latuszyński, K., Moores, M. T., & Stumpf-Fétizon, T. (2025). MCMC for multi-modal distributions. arXiv preprint arXiv:2501.05908.
> >
> > [6] Brooks, S., Gelman, A., Jones, G., & Meng, X. L. (Eds.). (2011). Handbook of markov chain monte carlo. CRC press.
> >
> > [7] Richter, L., Sallandt, L., & Nüsken, N. (2021, July). Solving high-dimensional parabolic PDEs using the tensor train format. In International Conference on Machine Learning (pp. 8998-9009). PMLR.
> >
> > [8] Eigel, M. and Gruhlke, R. and Marschall, M. (2022). Low-rank tensor reconstruction of concentrated densities with application to Bayesian inversion. Statistics and computing, Springer.
> >
> > [9] Strössner, C. and Sun, B. and Kressner, D. (2024) Approximation in the extended functional tensor train format. Advances in Computational Mathematics, Springer.
> >
> > [10] Bigoni, D. and Engsig-Karup, A. P. and Marzouk, Y. M. (2016). Spectral tensor-train decomposition. SIAM Journal on Scientific Computing.
> >
> > [11] Molozhavenko, A., and Maxim R. (2025) Optimization on the Extended Tensor-Train Manifold with Shared Factors. arXiv preprint arXiv:2508.20928
> >
> > [12] Zhang, Q., & Chen, Y. (2021). Path integral sampler: a stochastic control approach for sampling. arXiv preprint arXiv:2111.15141.

---

> ### Author Response · Authors · 2025-12-03
> **General response**
>
> We thank the reviewer again for their insightful and constructive comments. We have addressed these points in our updated manuscript and provided a high-level overview in our general response, 'Summary of contributions and revisions.'

---

### Author Response · Authors · 2025-12-03
**Summary of contributions and revisions**

Dear Area Chairs,

Thank you for taking the time to evaluate our paper. For your convenience, we provide a summary of the key contributions of our work, along with the discussion during the rebuttal phase and the resulting improvements. We appreciate the reviewers' insightful feedback, which allowed us to address important points and significantly strengthen our submission.

### Summary of our contributions

* **A diffusion-based sampler without neural networks**
We introduce the first diffusion-based sampling algorithm that replaces neural networks (NNs) with functional tensor trains (FTTs). FTTs are particularly advantageous when the target distribution admits a low-rank structure.
* **A BSDE-based, regression-driven training procedure**
By embedding the sampler in a BSDE framework, we leverage the structure of FTTs and develop a regression-based training algorithm. Avoiding potentially slow SGD-based optimization, this leads to substantially faster convergence. Although FTTs present their own numerical challenges, we address these through several algorithmic innovations, including appropriate basis constructions, outer-loop stabilization, dynamically moving domains along sample trajectories, adaptive regularization, and adaptive rank selection.
* **Strong empirical speed-accuracy performance**
Our experiments show that the proposed sampler achieves a highly favorable speed-accuracy trade-off: it can be both faster and more accurate than existing (diffusion-based) samplers. In particular, it significantly improves upon DIS and PIS - even though it uses essentially the same SDE model - while requiring fewer target evaluations (see Figures 3, 14, 15).
* **A first step toward FTT-based generative modeling**
We view this work as an initial step in bringing FTTs into diffusion-based sampling. Our results demonstrate that FTTs can yield faster and more robust algorithms, offering a promising alternative to the often fragile and slow optimization procedures required by neural networks in the context of sampling algorithms.


### Summary of revisions and main improvements

We thank the reviewers for their constructive feedback and are encouraged that they recognized the mathematical rigor and presentation of our work. Based on the critical points raised, we have significantly updated the manuscript. The major improvements are summarized below:
* **Expanded baseline comparisons:** We have added thorough comparisons against alternative sampling methods (Figure 14; see Section B.4). These results confirm that our method can also outperform many other state-of-the-art baselines, validating our claimed computation-accuracy tradeoff.
* **New experimental validation:** We have introduced three new experimental settings to demonstrate the method's versatility:
    * **Quantum physics:** A $\phi^4$ theory experiment involving highly constrained, correlated geometry in high dimensions (Section 4.2).
    * **Analytical validation:** An anisotropic Gaussian experiment that leverages analytical formulas to study rank behavior (Section 4.3).
    * **Statistical application:** A state-space model validation (Section B.5).
* **Hyperparameter robustness:** We have added a sensitivity analysis regarding key hyperparameters, demonstrating the robustness of our algorithm (see Figure 13).
* **Optimization alternatives:** We implemented Riemannian Gradient Descent (RGD) as an alternative to the baseline Alternating Least Squares (ALS). In terms of absolute computation time, and given the proximity of the initial values for the associated minimization problem, the block-wise optimization inherent to ALS significantly outperformed RGD. For this reason, we chose to focus exclusively on ALS in the updated manuscript and omit the discussion of RGD.
* **Clarification of mathematical contributions:** We have improved the readability of the introduction to Tensor Trains (TTs). Furthermore, we have clarified the necessary adjustments made to naive TT regressions, highlighting these adjustments as a distinct methodological contribution of our work (see Remark 3.3 and revisions throughout the text).
* **Enhanced adaptivity:** In the latest version of our proposed method, we introduced an additional adaptivity layer for the basis degrees, which further reduces computation time and mitigates error propagation. Details are provided in Appendix A.6.3.

Beyond these major changes, we have clarified individual questions and addressed minor misunderstandings in the detailed responses below.

---

### Meta-Review · Area_Chair_yBBV · 2025-12-17

**Summary:**

The paper proposes Tensor Train Diffusion (TTD), a solver for the Hamilton-Jacobi-Bellman (HJB) PDE underlying diffusion-based sampling. The key idea is to replace neural network approximations with Functional Tensor Trains (FTTs) and optimize them via a BSDE-based backward regression scheme using Alternating Least Squares (ALS).

While the reviewers acknowledged the novelty of introducing Tensor Trains to the diffusion sampling landscape and the mathematical rigor of the derivation, the paper falls short on empirical validation and applicability analysis. Despite the rebuttal, significant concerns regarding the method's practicality on complex, non-separable distributions remain.

**Reviewer Concerns:**

**Resolved**:

Transparency and Engineering Details (Reviewer Fub6): The authors addressed concerns regarding the "hidden" complexity of the method (e.g., adaptive regularization, basis selection) by moving these details to the main text.

Sensitivity Analysis (Reviewer Fub6, tHHw): The addition of hyperparameter sensitivity analyses was appreciated and addressed the reproducibility concerns regarding the tuning of the ALS solver.


**Outstanding**:

Limited Experimental Validation (Reviewers RsEA, tHHw, ZdeQ, Fub6 ): This is the critical bottleneck. The primary benchmark, the Multiwell potential, is known to be quasi-separable, making it a "best-case scenario" for Tensor Train decomposition. While the authors added Quantum and Gaussian experiments in the rebuttal, these setups still largely fall into regimes where low-rank structures are expected or mathematically enforced. The paper lacks demonstration on highly entangled high-dimensional tasks that would stress-test the FTT representation capabilities.

Insufficient Baselines (Reviewer RsEA): The comparison methods are insufficient. The paper primarily compares against other diffusion-based samplers (DIS, PIS) but lacks direct comparisons against established MCMC baselines (like HMC or NUTS) on the newly introduced tasks to benchmark wall-clock efficiency and accuracy.

Applicability and Low-Rank Assumption (Reviewers ZdeQ, Fub6 ): The method's success hinges entirely on the HJB solution admitting a low-rank approximation. The authors acknowledged this is an open question but did not provide sufficient theoretical or empirical evidence to characterize when this assumption holds. Specifically, the relationship between the required TT rank and the problem dimension (d) for non-trivial distributions remains unclear. Without this, it is difficult to define the boundaries of the algorithm's applicability.

**Reviewer Scores:**

**Reviewer RsEA** (2 -> 2): The reviewer’s fundamental concerns regarding the  comparisons to well-established workhorses such as NUTS, HMC, pSMC remain largely unaddressed. Besides, the reviewer’s fundamental concerns regarding  evaluation on standard benchmarks such as those used in Zhang & Chen (2022) also remain largely unaddressed.

**Reviewer tHHw:** (6-> 6): While initially positive, this reviewer flagged the "Core experimental results on Multiwell only" as a weakness. The rebuttal did not provide sufficiently complex "real-world" tasks to fully alleviate this concern.

**Reviewer ZdeQ:** (6-> 4): this reviewer flagged the "The experimental validation is limited, focusing exclusively on the Multiwell benchmark family". The rebuttal did not provide sufficiently complex "real-world" tasks to fully alleviate this concern. Besides, this reviewer asked for a characterization of how rank scales with dimension. The authors' response focused on parameter counting rather than the empirical growth of the rank itself for general problems, leaving the scalability concern open.

**Reviewer Fub6:** (2-> 2): The authors did an excellent job addressing the specific requests for transparency (limitations section, sensitivity analysis). However, the reviewer's underlying point about the "strong low-rank assumption" remains a valid reason for rejection.

---

### Decision · Program_Chairs · 2026-01-26

Reject